# Worsening of the Toxic Effects of (±)*Cis*-4,4′-DMAR Following Its Co-Administration with (±)*Trans*-4,4′-DMAR: Neuro-Behavioural, Physiological, Immunohistochemical and Metabolic Studies in Mice

**DOI:** 10.3390/ijms22168771

**Published:** 2021-08-16

**Authors:** Micaela Tirri, Paolo Frisoni, Sabrine Bilel, Raffaella Arfè, Claudio Trapella, Anna Fantinati, Giorgia Corli, Beatrice Marchetti, Fabio De-Giorgio, Cristian Camuto, Monica Mazzarino, Rosa Maria Gaudio, Giovanni Serpelloni, Fabrizio Schifano, Francesco Botrè, Matteo Marti

**Affiliations:** 1LTTA Center and University Center of Gender Medicine, Department of Translational Medicine, Section of Legal Medicine, University of Ferrara, 44121 Ferrara, Italy; micaela.tirri@unife.it (M.T.); sabrine.bilel@unife.it (S.B.); raffaella.arfe@unife.it (R.A.); giorgia.corli@edu.unife.it (G.C.); beatrice.marchetti@edu.unife.it (B.M.); rosamaria.gaudio@unife.it (R.M.G.); 2Department of Medical Sciences, University of Ferrara, 44121 Ferrara, Italy; paolo.frisoni@unife.it; 3Department of Chemistry and Pharmaceutical Sciences, University of Ferrara, 44121 Ferrara, Italy; claudio.trapella@unife.it (C.T.); anna.fantinati@unife.it (A.F.); 4Department of Health Care Surveillance and Bioetics, Section of Legal Medicine, Università Cattolica del Sacro Cuore, 00168 Rome, Italy; fabio.degiorgio@unicatt.it; 5Fondazione Policlinico Universitario A. Gemelli IRCCS, 00168 Rome, Italy; 6Laboratorio Antidoping FMSI, Largo Giulio Onesti 1, 00197 Rome, Italy; cristian.camuto@uniroma1.it (C.C.); monica.mazzarino@gmail.com (M.M.); francesco.botre@uniroma1.it (F.B.); 7Neuroscience Clinical Center & TMS Unit, 37138 Verona, Italy; g.serpelloni@gmail.com; 8Department of Psychiatry in the College of Medicine, Drug Policy Institute, University of Florida, Gainesville, FL 32611, USA; 9Psychopharmacology, Drug Misuse and Novel Psychoactive Substances Research Unit, School of Life and Medical Sciences, University of Hertfordshire, Hatfield AL10 9AB, UK; f.schifano@herts.ac.uk; 10Institute of Sport Science, University of Lausanne (ISSUL), Synathlon, 1015 Lausanne, Switzerland; 11Collaborative Center for the Italian National Early Warning System, Department of Anti-Drug Policies, Presidency of the Council of Ministers, 00186 Rome, Italy

**Keywords:** 4-4′-DMAR, immunohistochemistry, drug metabolism, hyperthermia, novel psychoactive substances, stimulant, oxidative/nitrosative stress, apoptosis, neurotoxicity, cortex

## Abstract

4,4’-Dimethylaminorex (4,4’-DMAR) is a new synthetic stimulant, and only a little information has been made available so far regarding its pharmaco-toxicological effects. The aim of this study was to investigate the effects of the systemic administration of both the single (±)*cis* (0.1–60 mg/kg) and (±)*trans* (30 and 60 mg/kg) stereoisomers and their co-administration (e.g., (±)*cis* at 1, 10 or 60 mg/kg + (±)*trans* at 30 mg/kg) in mice. Moreover, we investigated the effect of 4,4′-DMAR on the expression of markers of oxidative/nitrosative stress (8-OHdG, iNOS, NT and NOX2), apoptosis (Smac/DIABLO and NF-κB), and heat shock proteins (HSP27, HSP70, HSP90) in the cerebral cortex. Our study demonstrated that the (±)*cis* stereoisomer dose-dependently induced psychomotor agitation, sweating, salivation, hyperthermia, stimulated aggression, convulsions and death. Conversely, the (±)*trans* stereoisomer was ineffective whilst the stereoisomers’ co-administration resulted in a worsening of the toxic (±)*cis* stereoisomer effects. This trend of responses was confirmed by immunohistochemical analysis on the cortex. Finally, we investigated the potentially toxic effects of stereoisomer co-administration by studying urinary excretion. The excretion study showed that the (±)*trans* stereoisomer reduced the metabolism of the (±)*cis* form and increased its amount in the urine, possibly reflecting its increased plasma levels and, therefore, the worsening of its toxicity.

## 1. Introduction

Synthetic stimulants are one of the largest classes of novel psychoactive substances (NPS) seized and identified each year by law enforcement and forensic laboratories [1]. Among the latter, in recent years 4,4′-dimethylaminorex (4-Methyl-5-(4-methylphenyl)-4,5-dihydrooxazol-2-amine), commonly known as 4,4′-DMAR, has been both seized in various European countries and associated with cases of poisoning and deaths [2,3,4,5].

4,4′-DMAR may be considered a methyl-derivative of 4-methylaminorex (4-MAR) and aminorex [6], two synthetic stimulants known for their anorexic properties [7]. Indeed, in 4-MAR the methyl group is in position 4 on the oxazoline ring, such as 4,4′-DMAR that has one more methyl group in para-position on the phenyl ring (Figure 1).

Aminorex, on the other hand, has no methyl group on the oxazoline and phenyl rings. The presence of two chiral centres on the oxazoline ring of 4-MAR and 4,4-DMAR gives rise to four enantiomers represented by two diastereomeric (±)*-cis* and (±)*-trans* forms [3].

4,4′-DMAR and 4-MAR are widely used as recreational substances, as was aminorex [8]; due to their toxicity, all three stimulants are already under national or international legislation control.

As with other NPS, despite legislative restrictions 4,4′-DMAR can be purchased through websites selling “research chemicals” where it is usually sold in different forms (e.g., as white powder or as different coloured tablets and pellets) and under different names, including: “Serotoni”, “4-methyl-U4Euth” or “4-methyl-euphoria” [5,9,10,11,12]. The routes of administration mostly used by consumers include nasal insufflation, inhalation and oral administration [12]. Nevertheless, according to a fatality report from Hungary, 4,4′-DMAR can also be injected [9]. Low dosages (e.g., 10–15 mg for insufflation or 10–25 mg for oral administration) are usually ingested to enjoy levels of hyper-arousal, euphoria, decreased appetite and motor stimulation. An intermediate dosage (e.g., 20–50 mg oral) can be associated with restlessness, agitation and insomnia; high dosages (e.g., above 100 mg) may present with severe anorexia, mild paranoia at times with hallucinations, hyperthermia, bruxism, facial spasms, an increase in aggression levels, seizures and increased heart rate levels, at times evolving in a cardiac arrest [6,10,13]. However, these dose-related pharmaco-toxicological reports need to be interpreted with caution, since deriving from subjective users’ experiences and/or based on clinical descriptions of clients presenting to accident and emergency departments often referring to synthetic stimulant multi-drug intake episodes [10]. Indeed, in all the 27 4,4′-DMAR-related deaths reported to the EMCCDA, at least one further recreational drug and/or stimulant NPS was detected [9].

In vitro studies demonstrated that 4,4′-DMAR may elicit a potent inhibitory activity on dopamine (DAT), noradrenaline (NET) and serotonin transporters [8,14,15]. Moreover, (±)*cis*-4,4′-DMAR also inhibited vesicular monoamine transporter 2 (VMAT2) at a potency similar to 3,4-methylenedioxymethylamphetamine (MDMA), demonstrating that 4,4′-DMAR is a potent, non-selective, monoamine-releasing agent [15].

In a recent study, 4,4′-DMAR showed a DAT/SERT ratio of 0.4 [8], and the authors suggested that 4,4′-DMAR was also the most potent releaser of 5-HT from rat brain synaptosomes compared with d-amphetamine, aminorex and 4-MAR [3]. Furthermore, 4,4′-DMAR binds with relatively low affinity at both the 5-HT_2A_ (Ki~8.9 μM) and 5-HT_2C_ (Ki~11.0 μM) receptors [15], suggesting mild hallucinogenic effects [8]. Overall, one could conclude that 4,4′-DMAR may have a pharmaco-toxicological serotonergic profile similar to that of MDMA.

Cardiac arrest, brain oedema, elevated body temperature, bleeding and seizures were the most common adverse events/autopsy findings in fatalities associated with 4,4′-DMAR [9]. The admission notes and autopsy reports of 4,4′-DMAR-related deaths [9] identified a range of clinical features consistent with serotonin toxicity [16]. Conversely, the 4,4′-DMAR-related cardiotoxicity may be related to its effects on both extracellular norepinephrine and 5-HT [3,17].

Nevertheless, there are no published pre-clinical or scientific safety data relating to the toxic/neurotoxic potential of 4,4′-DMAR in animals or humans. A further problem is represented by the fact that in the seized material, the exact concentration of either (±)*cis*-4,4′-DMAR or of the (±)*trans-*4,4′-DMAR stereoisomer, or indeed of their presence in a racemic mixture, is not known [18]. This issue, already previously described for stimulant compounds in racemic mixtures such as threo-methylphenidate [19], (±)-MDPV [20,21] or empathogenic drugs such as (±)-MDMA [22], could directly influence these molecules’ pharmaco-toxicological profile, potentially representing a serious health problem for consumers but also complicating data interpretation in forensic toxicology analysis.

Therefore, the aim of the present study was to investigate the effects of the acute systemic administration of both the single (±)*cis* (0.1–60 mg/kg) and (±)*trans* (30 and 60 mg/kg) stereoisomers but also of their co-administration ((±)*cis* at 1, 10 or 60 mg/kg + (±)*trans* at 30 mg/kg) in mice. A range of physiological (e.g., sweating, salivation, hyperthermia) and neuro-behavioural (e.g., psychomotor agitation, aggressiveness, convulsion) parameters were here assessed. Moreover, to highlight possible neurotoxic mechanisms, and similar to what previously investigated in relation to MDMA [23] the effects of 4,4′-DMAR on the expression of key markers of oxidative/nitrosative stress (8-OHdG, iNOS, NT and NOX2) and apoptosis (Smac/DIABLO and NF-κB) were here analysed. Furthermore, in taking from some current preliminary data, which showed the emergence of a 4,4′-DMAR-related hyperthermia, the expression of heat shock proteins (HSP27, HSP70, HSP90), markers related to heat-induced response [24,25,26], was here assessed as well. Finally, in studying the excretion of the related urinary metabolites, the potential mechanism underlying the potentiation of the toxic effects of the (±)*cis* stereoisomer when co-administered with the (±)*trans* stereoisomer was here investigated.

## 2. Results

### 2.1. Behavioural Studies

Overall, the systemic administration of (±)*cis*-4,4′-DMAR (e.g., 1–60 mg/kg i.p.), (±)*trans*-4,4′-DMAR (e.g., 30 and 60 mg/kg), and of both together (e.g., 1–10–60 mg/kg (±)*cis*-4,4′-DMAR + 30 mg/kg (±)*trans*-4,4′-DMAR i.p.) was here associated with important physiological and neuro-behavioural changes in mice.

#### 2.1.1. Physiological and Neuro-Behavioural Responses with (±)*Cis*-4,4′-DMAR

(±)*Cis*-4,4′-DMAR (0.1–60 mg/kg i.p.; Table 1) induced levels of psychomotor agitation in mice. The effect was significant in the dose range of 3–60 mg/kg; both the latency of the appearance of the effect (one-way ANOVA; *F* (3.20) = 66.56, *p* < 0.0001) and its duration (one-way ANOVA showed *F* (3.20) = 66.56, *p* < 0.0001) were dose-dependent.

All animals showed profuse sweating and salivation at the doses of 30 and 60 mg/kg, although at the highest dose of 60 mg/kg only 50% of the animals showed profuse sweating possibly because at this dose half of them died before clearly showing this physiological effect. (±)*Cis*-4,4′-DMAR dose-dependently induced hyperthermia at 30 mg/kg (+1.52 ± 0.11 °C) and at 60 mg/kg (+2.22 ± 0.12 °C), whilst lower doses (0.1–10 mg/kg) were ineffective (one-way ANOVA *F* (6.35) = 52.18, *p* < 0.0001). (±)*Cis*-4,4′-DMAR was not associated with increased levels of *spontaneous aggressiveness* in mice. Conversely, the molecule induced levels of *stimulated aggressiveness* in a dose-dependent manner at 10 and 30 mg/kg (*t* =12.30, *df* = 10; *p* < 0.0001). At the highest dose (60 mg/kg), the spontaneous and stimulated aggressiveness tests could not be performed as the animals were excessively agitated and convulsive. Systemic administration of (±)*cis*-4,4′-DMAR induced seizures in 50% and 100% of mice at the 30 and 60 mg/kg dosages, respectively; the duration of seizures was longer in those mice treated with the highest dose (*t* = 3.280, *df* = 10; *p* = 0.0083). (±)*Cis*-4,4′-DMAR at 30 and 60 mg/kg caused the death of 50% and 100% of mice, respectively, and at the highest dose the time to death was significantly shorter (*t* = 2.526, *df* = 10; *p* = 0.0301).

#### 2.1.2. Physiological and Neuro-Behavioural Responses to (±)*Trans*-4,4′-DMAR

Systemic administration of (±)*trans*-4,4′-DMAR (30 and 60 mg/kg; Table 2) did not induce any physiological and neuro-behavioural changes at either of the dose tested. The (±)*trans*-4,4′-DMAR tested at the lower doses of 0.1 and 1 mg/kg was inactive on all parameters studied.

#### 2.1.3. Physiological and Neuro-Behavioural Responses to the Concurrent (±)*Cis*-4,4′-DMAR + (±)*Trans*-4,4′-DMAR Administration

Co-administration of increasing doses of (±)*cis*-4,4′-DMAR (e.g., 1, 10 and 60 mg/kg) with (±)*trans*-4,4′-DMAR (e.g., 30 mg/kg) worsened the physiological and neuro-behavioural alterations caused by single injections of the stereoisomer (±)*cis*-4,4′-DMAR (e.g., 1, 10 and 60 mg/kg; Table 3).

The first behavioural parameter affected by co-administration was the psychomotor agitation. (±)*Cis*-4,4′-DMAR at 1 mg/kg, ineffective alone, when co-administered with (±)*trans*-4,4′-DMAR at 30 mg/kg induced psychomotor agitation in 50% of the treated mice with an effect duration of approximately one hour. At 10 mg/kg (±)*cis*-4,4′-DMAR alone caused psychomotor agitation in 100% of the treated mice with an effect that started after about thirty minutes and lasted for about two hours; conversely, when co-injected with (±)*trans*-4,4′-DMAR the stimulatory effect was anticipated, e.g., it began about four minutes after the drug administration and persisted for about three hours (*t* = 3.900, *df* = 10; *p* = 0.030). Moreover, co-administration of (±)*cis*-4,4′-DMAR at 10 mg/kg with (±)*trans*-4,4′-DMAR at 30 mg/kg induced sweating and salivation in 33% of the animals, hyperthermia (Δ°C~1.6 °C; *t* = 11.80, *df* = 10; *p* < 0.0001) and enhanced stimulated aggressiveness measured as number of bites (*t*= 16.99, *df* = 10; *p* < 0.0001). Co-administration of the highest dose of (±)*cis*-4,4′-DMAR (60 mg/kg) with (±)*trans*-4,4′-DMAR at 30 mg/kg promoted salivation in 50% of mice, caused hyperthermia (Δ°C~2.25 °C), increased the duration of seizure episodes (*t* = 2.996, *df* = 10; *p* = 0.0134) whilst reducing the number of episodes (*t* = 2.439, *df* = 10; *p* = 0.0349), and caused more rapidly the death of mice (*t* = 3.063, *df* = 10; *p* = 0.0120).

### 2.2. Results of Tissue Samples Collection

#### 2.2.1. Histological Results

Histological analyses showed only a slight oedema in the cerebral cortex (Figure 2).

#### 2.2.2. Immunohistochemical Results

Systemic administration of (±)*cis*-4-4′-DMAR (10–60 mg/kg) increased the expression of markers of oxidative/nitrosative stress (8-OHdG, iNOS, and NT), apoptosis (Smac/DIABLO and NF-κB) and heat shock proteins (HSP27 and HSP70) in the frontal cortex of mice. Moreover, the co-administration of (±)*cis*-4-4′-DMAR (10 and 60 mg/kg) with an ineffective dose of (±)*trans*-4-4′-DMAR (30 mg/kg) enhanced the hyperexpression of iNOS, Smac/DIABLO and HSP27 demonstrating the increased action of (±)*cis* and (±)*trans* co-administration.

Systemic administration of (±)*cis*-4-4′-DMAR at 30 and 60 mg/kg increased the immunoreactivity of 8-OHdG in the frontal cortex of mice (one-way ANOVA; *F* (3.30) = 3.507, *p* = 0.0272; Figure 3A).

Co-administration of (±)*cis*-4-4′-DMAR (at 10 and 60 mg/kg), with the ineffective dose of (±)*trans*-4-4′-DMAR at 30 mg/kg did not modify the expression of 8-OHdG in the frontal cortex with respect to that induced by the single injection of (±)*cis*-4-4′-DMAR (one-way ANOVA showed F (5.44) = 4.109, *p* = 0.0038; Figure 3A).

Systemic administration of (±)*cis*-4-4′-DMAR at 60 mg/kg increased the immunoreactivity of iNOS in the frontal cortex of mice (one-way ANOVA showed *F* (3.30) = 87.32, *p* < 0.0001; Figure 3B). The effect of (±)*cis*-4-4′-DMAR at 60 mg/kg was enhanced when co-injected with the ineffective dose of (±)*trans*-4-4′-DMAR at 30 mg/kg (one-way ANOVA showed *F* (5.44) = 4.109, *p* = 0.0038; Figure 3B). Co-injection of (±)*cis*-4-4′-DMAR at 10 mg/kg with (±)*trans*-4-4′-DMAR at 30 mg/kg did not overexpress the immunoreactivity of iNOS.

Administration of (±)*cis*-4-4′-DMAR reduced at 10 mg/kg and increased at 30 mg/kg the immunoreactivity of 3-nitrotyrosine (NT) expression in the frontal cortex of mice (one-way ANOVA showed *F* (3.30) = 25.64, *p* < 0.0001; Figure 3C). At the highest dose of 60 mg/kg (±)*cis*-4-4′-DMAR did not affect NT expression. The effect of (±)*cis*-4-4′-DMAR at 60 mg/kg was enhanced when co-injected with the ineffective dose of (±)*trans*-4-4′-DMAR at 30 mg/kg (one-way ANOVA showed *F* (5.44) = 16.23, *p* < 0.0001; Figure 3C). Co-injection of (±)*cis*-4-4′-DMAR at 10 mg/kg with (±)*trans*-4-4′-DMAR at 30 mg/kg did not change the immunoreactivity of NT.

Administration of (±)*cis*-4-4′-DMAR reduced at 10 and 30 mg/kg the immunoreactivity of NOX-2 expression in the frontal cortex of mice (one-way ANOVA showed *F* (3.30) = 6.179, *p* = 0.0038; Figure 3D). At the highest dose of 60 mg/kg (±)*cis*-4-4′-DMAR did not affect NOX-2 expression. Co-injection of (±)*cis*-4-4′-DMAR at 10 and 60 mg/kg with (±)*trans*-4-4′-DMAR at 30 mg/kg did not change the immunoreactivity of NOX-2 with respect to that induced by the single injection of (±)*cis*-4-4′-DMAR (one-way ANOVA; *F* (5.44) = 3.935, *p* = 0.0051; Figure 3D).

Systemic administration of (±)*cis*-4-4′-DMAR at 30 and 60 mg/kg increased the immunoreactivity of Smac/DIABLO in the frontal cortex of mice (one-way ANOVA; *F* (3.30) = 49.50, *p* < 0.0001; Figure 4A).

The effect of (±)*cis*-4-4′-DMAR at 60 mg/kg was enhanced when co-injected with the ineffective dose of (±)*trans*-4-4′-DMAR at 30 mg/kg (one-way ANOVA; *F* (5.44) = 51.94, *p* < 0.0001; Figure 4A). Co-injection of (±)*cis*-4-4′-DMAR at 10 mg/kg with (±)*trans*-4-4′-DMAR at 30 mg/kg did not overexpress the immunoreactivity of Smac/DIABLO.

Administration of (±)*cis*-4-4′-DMAR at 10, 30 and 60 mg/kg increased the immunoreactivity of NF-kB in the frontal cortex of mice (one-way ANOVA; *F* (3.30) = 49.50, *p* < 0.0001; Figure 4B). The effect of (±)*cis*-4-4′-DMAR at 10 and 60 mg/kg was not changed when co-injected with the ineffective dose of (±)*trans*-4-4′-DMAR at 30 mg/kg (one-way ANOVA; *F* (5.44) = 51.94, *p* < 0.0001; Figure 4B).

(±)*Cis*-4-4′-DMAR at 30 and 60 mg/kg increased the immunoreactivity of HSP27 in the frontal cortex of mice (one-way ANOVA showed *F* (3.30) = 55.12, *p* < 0.0001; Figure 5A), whilst (±)*cis*-4-4′-DMAR at 10 mg/kg, alone, was ineffective. Co-injection of (±)*cis*-4-4′-DMAR at 10 and 60 mg/kg with the ineffective dose of (±)*trans*-4-4′-DMAR at 30 mg/kg overexpressed the immunoreactivity of HSP27 with respect to that induced by the single injection of (±)*cis*-4-4′-DMAR (one-way ANOVA; *F* (5.44) = 63.85, *p* < 0.0001; Figure 5A).

(±)*Cis*-4-4′-DMAR at 30 and 60 mg/kg increased the immunoreactivity of HSP70 in the frontal cortex of mice (one-way ANOVA; *F* (3.30) = 73.77, *p* < 0.0001; Figure 5B), whilst (±)*cis*-4-4′-DMAR at 10 mg/kg, alone, was ineffective. Co-injection of (±)*cis*-4-4′-DMAR at 10 mg/kg with the ineffective dose of (±)*trans*-4-4′-DMAR at 30 mg/kg overexpressed the immunoreactivity of HSP70 respect to that induced by the single injection of (±)*cis*-4-4′-DMAR (one-way ANOVA; *F* (5.44) = 26.72, *p* < 0.0001; Figure 5B). Conversely, the co-injection of (±)*cis*-4-4′-DMAR at 60 mg/kg with the ineffective dose of (±)*trans*-4-4′-DMAR at 30 mg/kg increased the expression of the immunoreactivity of HSP70 similarly to that induced by the single injection of (±)*cis*-4-4′-DMAR at 60 mg/kg.

(±)*Cis*-4-4′-DMAR reduced at 10 and 30 mg/kg the immunoreactivity of HSP90 expression in the frontal cortex of mice (one-way ANOVA; *F* (3.30) = 9.819, *p* = 0.0001; Figure 5C). The (±)*trans*-4-4′-DMAR at 30 mg/kg reduced the expression of HSP90., whilst the co-injection of (±)*cis*-4-4′-DMAR at 10 and 60 mg/kg with (±)*trans*-4-4′-DMAR at 30 mg/kg did not change the immunoreactivity of HSP90 with respect to that induced by the single injection of (±)*cis*-4-4′-DMAR (one-way ANOVA; *F* (5.44) = 9.040, *p* < 0.0001; Figure 5C).

### 2.3. Excretion Studies on Urine Samples

The principal markers of intake for (±)*cis*-4,4′-DMAR were the carboxylate (M2) and mono-hydroxylate (M1, M3) metabolites where M2 was the most excreted metabolite detected in mice urine samples, while the (±)*trans*-4,4′-DMAR did not exhibit phase I/II metabolism, see Figure 6 for the structures of metabolites.

In the present study, the excretion levels of M1, M2 and M3 were compared in the two groups of mice. The first group was administered with (±)*cis* isomer and the second group was co-administered with both isomers at the same dose (10 mg/kg). The aim was to evaluate changes in the excretion profile of these metabolites (M1–M3) between the two groups. The excretion levels of the individual metabolites were compared as a percentage ratio. This ratio was calculated considering the excretion of the parent compound, assuming the percentage of (±)*cis*-4,4′-DMAR as 100%. The data were graphically reported as box plots, with the calculated minimum, maximum, median and average values (Figure 7A).

The values shown were the sum of the excretion of (±)*cis*-4,4′-DMAR and its metabolites in the range of hours considered. The results showed that the administration of (±)*cis*-4,4′-DMAR alone led to the excretion of M1 and in particular of M2 in greater quantities than the parent compound. Furthermore, a significant variability of the data was observed for all the metabolites, which could be attributed to the variable nature of the excretion among different mice and time ranges. The results were significantly different in the case of co-administration of both isomers. For all the metabolites there was a remarkable reduction in excretion values compared to the parent compound, mainly for the metabolites M1 and M2. The principal marker M2 showed maximum percentage values equal to half of the excretion of the parent compound whilst the average values of all three metabolites were significantly lower than this value. This observation could be explained by an inhibition of the metabolism of the (±)*cis* isomer caused by the co-administration with the (±)*trans* isomer. The relatively low variability of the data could be explained by the reduced metabolism of the parent compound, which entailed a lower excretion of the metabolites and consequently smaller levels of inter-individual variability. This was also highlighted by the comparison of excretion data showed in Figure 7B with data normalised to the sum of excretion of 4,4′-DMAR and its metabolites.

### 2.4. Physiological and Neuro-Behavioural Responses with (±)Cis-4,4′-DMAR + (±)Trans-4,4′-DMAR Co-Administration

Animals from which urine for excretion studies were obtained (Figure 7) were simultaneously observed to see if changes in the excretion metabolism of the (±)cis-4,4′-DMAR stereoisomer were associated with changes in physiological and neuro-behavioural responses (Table 4).

Co-administration of (±)*cis*-4,4′-DMAR at 10 mg/kg with (±)*trans*-4,4′-DMAR at 10 mg/kg worsened the physiological and neuro-behavioural alterations caused by single injections of the stereoisomer (±)*cis*-4,4′-DMAR. In particular, (±)*cis*-4,4′-DMAR when co-administered with (±)*trans*-4,4′-DMAR reduced the time of onset of psychomotor agitation (*t* = 2.434, *df* = 10; *p* = 0.0352) and prolonged its duration (*t* = 3.355, *df* = 10; *p* = 0.0073); furthermore, it promoted hyperthermia and enhanced stimulated aggressiveness measured as number of bites (*t* = 11.36, *df* = 10; *p* < 0.0001) in mice.

## 3. Discussion

The present study demonstrated for the first time the toxicity of the (±)*cis*-4,4′-DMAR in mice and the risk when taken together with the (±)*trans*-4,4′-DMAR form in racemic mixtures. (±)*Cis*-4,4′-DMAR dose-dependently induced psychomotor agitation, sweating, salivation, hyperthermia, stimulated aggression, convulsions and death in mice. Conversely, the (±)*trans*-4,4′-DMAR was inactive. However, their co-administration resulted in a worsening of the toxic effects caused by (±)*cis*-4,4′-DMAR, inducing rapid and severe hyperthermia, convulsions and death of the animals. Immunohistochemical analysis showed that this acute intoxication caused high expression of markers of oxidative/nitrosative stress (8-OHdG, iNOS, NT and NOX2), apoptosis (Smac/DIABLO and NF-κB) and heat shock proteins (HSP27, HSP70, HSP90) in the frontal cortex, suggesting potential neurotoxic damage. The urinary excretion studies suggested that the worsening of physiological and neuro-behavioural parameters could be related to the inhibition of the metabolism of the (±)*cis*-4,4′-DMAR form by the (±)*trans*-4,4′-DMAR. Overall, the strength of the present study related here to its clinical–toxicological relevance of the use of doses of 4-4’-DMAR in mice, which are equivalent to those used in humans (HED; human equivalent dose); these were associated with mild, intermediate and strong responses (Table 5). Although the current study was solely based on the preclinical mouse model, this may allow a translational evaluation of the pharmaco-toxicological effects that could be observed in humans.

Physiological and neuro-behavioural alterations observed in mice were broadly in agreement with the clinical scenario typically observed in consumers ingesting 4-4’-DMAR, with symptoms and signs including: psychomotor agitation, sweating, salivation, hyperthermia, aggression, convulsions and possible death [4,10,13]. This scenario is likely related to the increased release of catecholamines and facilitated serotonergic transmission [6]. In particular, the latter seems to be responsible for the more dangerous toxic effects found in 4-4’-DMAR intoxications, such as cardiac arrest, brain oedema, hyperthermia, bleeding and seizures [4]. Therefore, this study allows us to hypothesise that from a clinical point of view an acute 4-4’-DMAR intoxication can be treated, such as that caused by other stimulants already known. Furthermore, the metabolic profile of 4-4’-DMAR excretion can cover an important aspect in clinical–toxicological and forensic investigations. While changes in immunohistochemical markers (for example HSPs) are indicative of metabolic changes in the brain that may be of help in forensic investigations to clarify, for example, the presence of hyperthermia in the brain and its possible pathophysiological relapse.

### 3.1. Psychomotor Agitation

Similarly to 4-MAR, psychomotor agitation was the first behavioural effect observed after administration of (±)*cis*-4,4′-DMAR; not only an increase in spontaneous locomotion activity [27], but also an “unusual hyperactivity” pattern (e.g., rapid and sudden horizontal displacements in all directions, sustained turning behaviour, involuntary falls from the high plate during the evaluation of the tests, stereotyped movement and rearing activity) were observed [28]. These enhanced behavioural responses were consistent with an increase in extracellular levels of monoamines [28,29,30], especially in the dorsal and ventral striatum [31,32,33].

Turning behaviour and stereotypies induced by 4-4′-DMAR were typically reported after the administration of other synthetic stimulants, such as amphetamine, 4-MAR, MDPV, mephedrone, methedrone, α-PVP, α-PBP, 4′-MePPP, MDMA and MPA [28,34,35,36,37,38,39,40,41].

Although a previous study showed that the motor stimulation and stereotypies caused by 4-MAR are serotonin-independent and prevented only by the blockade of dopaminergic D_1_ and D_2_ receptors [28], the motor effects induced by 4-4′-DMAR could also depend on the serotonergic signal, given its pharmacodynamic profile more similar to that of MDMA with respect to 4-MAR and aminorex [3,14].

### 3.2. Aggression

(±)*Cis*-4,4′-DMAR caused psychomotor agitation associated with stimulated aggressiveness in mice as typically reported for other psychostimulants such as cocaine, MDPV, methiopropamine, methamphetamine, amphetamine [42,43,44], which well identifies the adverse behavioural effects typically observed in users of synthetic stimulants and in particular of cathinone that gives rise to the so-called excited delirium syndrome (EDS) [45,46].

The EDS was first described as the sudden, unexpected death of people restrained by law enforcement officers [47] (in particular, positional holds are the restraints most frequently associated with sudden death in susceptible people with excited delirium) [48]. Frequently, the subjects tend to display signs of agitation, aggressiveness and hyperactivity prior to the exitus [49].

### 3.3. Hyperthermia, Sweating, Salivation

One of the main physiological alterations caused by (±)*cis*-4,4′-DMAR was here hyperthermia. Psychostimulants may induce both hyperthermic and hypothermic responses in rodents depending on environmental temperature, psycho-physiological activation (stress-induced hyperthermia), drug pharmacodynamics, social interactions and gender [50,51,52]. In our experimental conditions (room temperature 22–23 °C), (±)*cis*-4,4′-DMAR induced hyperthermia, consistent with observations made with remaining synthetic stimulants such as MDMA, methamphetamine, cocaine, methylone, MDPV, α-PVP, PMMA and MDAI [30,50,53,54,55,56,57,58,59,60].

Hyperthermia, which is one of the symptoms of the serotonin syndrome [61], may be sustained by an increased serotonergic signal. In fact, a direct link between the serotonergic transmission and thermoregulation has been well established. (±)*Cis*-4,4′-DMAR, promoting serotonergic transmission and possibly activating 5HT_2A/C_ receptors [15], could induce hyperthermia by increasing metabolism whilst inducing hyperactivity coupled with hyperthermia [53,54] and by causing peripheral vasoconstriction [62,63]; this scenario has been typically reported as well for MDMA, cocaine, methylone, MDPV and 25B-NBOMe [51,56,57,64]. However, the 4,4′-DMAR-induced norepinephrine release may also account for its hyperthermic effect. Furthermore, increased norepinephrine plasma levels may lead to a loss of heat dissipation through α1AR-mediated vasoconstriction, while stimulation of α1 and β3 adrenergic receptors may regulate a mitochondrial protein in skeletal muscle, uncoupling protein-3 (UCP-3) inducing thermogenesis [65,66]. Overall, hyperthermia is considered a potential acute severe adverse effect and one of the primary causes of death [67,68]. According to the literature, this condition can be associated with a range of life-threatening complications, namely cerebral oedema, rhabdomyolysis, hyponatraemia, disseminated intravascular coagulation (DIC) and coma [69].

At present, knowledge of the effects of drug-induced hyperthermia on neurotoxicity is very limited. As reported by Bowyer at al., amphetamine- and methamphetamine-induced hyperthermia potentially enhances neurotoxicity through both the disruption of protein function, ion channels and enhanced ROS production and through its effects on the vasculature [70]. The induced hyperthermia may lead to transitory breakdowns in the blood–brain barrier, which result in neurodegeneration and neuroinflammation in laboratory animals and brain pathology [70,71]. In a study by Zhou et al., it was reported that the toxicity of 4-chloromethcathinone (4-CMC) and 4-methylmethcathinone (4-MMC) increased with changes in temperature, from 37 to 40.5 °C. According to the authors, the mechanism of mitochondrial toxicity of 4-CMC and 4-MMC, which is increased by hyperthermia, contributes to the neural toxicity of these compounds [72].

(±)*Cis*-4,4′-DMAR-induced hyperthermia was here associated by profuse sweating and salivation in mice, which are typical symptoms of the serotonin syndrome [61] being observed as well with other psychoactive substances in rodents such as MDAI [60], PMMA [59] and p-methoxy phenylethylamine [73].

### 3.4. Convulsions and Lethality

Seizures are among the most dangerous adverse effects caused by psychostimulants in humans [74,75]. Systemic administration of (±)*cis*-4,4′-DMAR caused convulsions in mice as evidenced by other stimulants such as cocaine [75,76,77,78,79,80], MDMA [81,82], methamphetamine and 4-methylaminorex [82].

In particular, systemic injection of (±)*cis*-4,4′-DMAR dose-dependently induced convulsions and death in adult male mice at 30 mg/kg, with the lethality of about 50%, and at 60 mg/kg, with the lethality of about 100%, showing that (±)*cis*-4,4′-DMAR similar to cocaine in inducing seizures and lethality [76,78] in mice. The pro-convulsive properties of (±)*cis*-4,4′-DMAR could be due to its greater affinity and blocking activity of monoamine and SERT transporters [3,8,14,15,29,83].

Surprisingly, in the present study, the (±)*trans*-4,4′-DMAR from a neuro-behavioural and physiological point of view was not active per se but was able to enhance the effects of (±)*cis*-4,4′-DMAR and increased its toxicity. At present, no in vivo pharmacological studies have been performed with the (±)*trans*-4,4′-DMAR stereoisomer whilst only one in vitro pharmacodynamic study has been published showing that (±)*trans*-4,4′-DMAR is a substrate-type releaser at the DAT, NET transporters and a SERT uptake blocker in rat brain tissue with a lower potency than the (±)*cis*-4,4′-DMAR stereoisomer [14]. However, in a recent pharmacokinetic study it was found that whilst the (±)*cis*-4,4′-DMAR compound is metabolised and excreted in the mouse urine, the (±)*trans*-4,4′-DMAR stereoisomer is not metabolised [84] suggesting a stereoselective metabolism and disposition of 4,4′-DMAR. This is common in stereoisomers of a drug that generally metabolised with different metabolic profile [85] both in pharmaceutical [86] and in abusing drugs [87,88].

Furthermore, the (±)*trans*-4,4′-DMAR may have a different binding to plasma or tissue proteins and transporters that is common in chiral drugs [89]. Its lack of pharmacological activity could, therefore, be due to the low availability of the compound at the central level. Further studies will need to be undertaken to investigate this aspect. However, the present study demonstrated that the (±)*trans*-4,4′-DMAR, although inactive from a neuro-behavioural and physiological point of view, caused a lower urinary excretion of metabolites (M1 and M2) of the (±)*cis*-4,4′-DMAR compound, causing an increase in the (±)*cis*-4,4′-DMAR itself (Figure 7B). This results of inhibition after co-administration of a mixture of both stereoisomers was in accordance with those obtained in similar studies [88,90,91]. This may suggest that the (±)*trans*-4,4′-DMAR could inhibit the (±)*cis*-4,4′-DMAR metabolism, and thus increasing its bioavailability levels. This hypothesis is consistent with the worsening of the physiological and neuro-behavioural alterations observed in the same animals from which the urinary excretion studies were made (Table 4). The worsening of the pharmaco-toxicological effects resulting from the co-administration of (±)*cis*-4,4′-DMAR and (±)*trans*-4,4′-DMAR was dramatic (Table 3) and demonstrated the great danger of taking these stereoisomers in mixture, as unfortunately could well occur with preparations purchased from rogue web sites. The availability of mixture of two chiral compounds to obtain an enhancement of the psychoactive effects has already been described and observed for other NPS, in particular with synthetic cannabinoids [92].

### 3.5. Immunoistochemical Studies

This was the first study to investigate the 4,4′-DMAR-induced brain damage. The results of in vivo studies suggested the occurrence of phenomena related to CNS alterations (e.g., hyperthermia; seizures). By analogy to other stimulant drugs (e.g., MDMA [23] and cocaine [93]) we hypothesised that the oxidative and/or nitrosative stress occurred in the brain following the intake of 4,4′-DMAR.

Hence, we investigated the expression of 8-OHdG, a marker of ROS-induced oxidative damage to DNA [94]. We observed that 8-OHdG was overexpressed in mice treated with 4-4′-DMAR, and especially so at higher dosages ((±)*cis* 30, (±)*cis* 60 and (±)*trans* 30 + (±)*cis* 60). These findings may well suggest that the administration of 4-4′-DMAR caused here levels of ROS-mediated DNA damage. We, therefore, evaluated the expression of NOX2, one of the main ROS-producing enzymes, which instead was not increased in treated mice compared to the controls. One could that conclude that in the mechanism of 4-4′-DMAR-induced ROS-mediated DNA damage other pathways are involved, which will need to be investigated in future studies. Among these, there could be the production of superoxide by iNOS, a phenomenon that can occur in conditions of oxidative stress [95]. This enzyme, which in the neurons is responsible for the production of a share of NO, under stress conditions can produce NO at a high rate, leading to the production of peroxynitrite, a very toxic oxidant and nitrating agent. We observed that iNOS expression in treated mice is greater than in controls, in particular at the dose of (±)*cis* 60 and (±)*trans* 30 + (±)*cis* 60, there is a large increase in iNOS expression (*p* < 0.001).

It was, therefore, possible to expect damage derived from reactive nitrogen species (RNS). So, we investigated the expression of NT, known to be a direct marker of nitrosative stress, being the product of tyrosine nitration mediated by RNS such as peroxynitrite anion and nitrogen dioxide, formed in the presence of NO [96]. We observed a statistically significant increase in NT at the (±)*cis* 30 and (±)*trans* 30 + (±)*cis* 60 doses, but not at the (±)*cis* 60 dose. At the dose of (±)*cis* 10, NT is even reduced compared to controls. From the analysis of these data, it is possible to hypothesise the occurrence of nitrosative stress following the administration of 4-4′-DMAR, however, it is not possible to draw conclusions on the effect this has on cellular DNA.

The results reported in the study were related to the analysis of the frontal cortex, however, from a preliminary analysis we did not detect changes in specific areas, but instead changes were widespread in the CNS regions examined (e.g., cortex, striatum, hippocampus, cerebellum).

Having observed the occurrence of hyperthermia in mice treated with 4-4′-DMAR, we investigated some heat shock proteins (HSP27 [24], HSP70 [25], HSP90 [26]), which have been shown to be correlated with thermal damage response. HSP27 was overexpressed in mice treated at dosages of (±)*cis* 30, (±)*cis* 60 and (±)*trans* 30 + (±)*cis* 60. Interestingly, there was a significant increase in HSP27 expression in mice treated with (±)*trans* 30 + (±)*cis* 10 and (±)*trans* 30 + (±)*cis* 60 compared to (±)*cis* 10 and (±)*cis* 60. This trend, which was consistent with the respective increments in temperature recorded in vivo, may suggest a greater potential of the (±)*cis*-(±)*trans* mixture in inducing hyperthermia and the consequent adaptive cellular response. HSP70 showed a similar trend, with the exception that the administration of (±)*trans* 30 + (±)*cis* 60 was not associated with HSP70 expression compared to (±)*cis* 60. One possible explanation is that HSP27 showed an early increase upon thermal stimulus compared to HSP70, and the mice treated with the (±)*cis* 60 assays all died rapidly, not allowing full expression of HSP70 [25]. Regarding HSP90, we observed a reduced expression in mice treated with 4-4′-DMAR. This trend is difficult to explain, as an increase in expression could reasonably have been expected, similarly to other HSPs. HSP90 is physiologically abundant within the cell and is important in the formation and function of several protein complexes that maintain cell homeostasis. In addition, it plays an important role in the thermal stress response [97]. Its hypo-expression could, therefore, reduce the cell’s ability to resist to 4-4′-DMAR-induced thermal and oxidative stress.

Thermic damage to cells can induce one of two opposing responses: apoptosis, that removes damaged cells to prevent inflammation and the heat shock response, to maintain cell survival. The modulation activity of apoptosis mechanisms exerted by HSPs is well known [98]. In particular, HSP70 acts on several targets, including JNK, preventing the release of cytochrome c from the mitochondria. JNK [99] is in turn involved in the release of Smac/DIABLO. HSP27 [100] also appeared to have an inhibitory role on the release of Smac/DIABLO. Indeed, Smac/DIABLO binds with the inhibitor of apoptosis proteins (IAPs), thus freeing caspases to activate apoptosis [101]. Our evaluation of Smac/DIABLO expression showed a significant increase at the doses of (±)*cis* 30, (±)*cis* 60 and (±)*trans* 30 + (±)*cis* 60. Furthermore, the immunoreactivity of Smac/DIABLO is greater in subjects treated with (±)*trans* 30 + (±)*cis* 60 compared to (±)*cis* 60. One could then conclude that high doses of 4-4′-DMAR have a pro-apoptotic effect. These data are consistent both with the trend of radical stress markers (in particular with 8-OHdG and iNOS), and with the expression of HSP27 and HSP70. It is not possible, with the current data, to determine whether 4-4′-DMAR induces oxidative/nitrosative stress per se or whether this was secondary to hyperthermia. These two phenomena are closely related considering, for example, that in conditions of hyperthermia, iNOS contributes to NO-dependent apoptosis [102].

ROS have been reported to both activate and repress NF-κB signalling. For instance, ROS often stimulates the NF-κB pathway in the cytoplasm but inhibits NF-κB activity in the nucleus [103]. In most cases the expression of NF-κB target genes typically promotes cellular survival, including those that antagonise the effects of ROS [104]. In light of this, our observation of NF-κB overexpression in 4-4′-DMAR-treated mice is not surprising. Regarding histological analyses, the finding of mild cerebral edema is consistent with what has been observed in fatalities associated with 4,4′-DMAR [9].

## 4. Materials and Methods

### 4.1. Animals

Eighty-four-male ICR (CD-1^®^) mice weighing 30–35 g (Centralised Preclinical Research Laboratory, University of Ferrara, Ferrara, Italy) were group housed (5 mice per cage; floor area per animal was 80 cm^2^; minimum enclosure height was 12 cm), exposed to a 12:12-h light–dark cycle (light period from 6:30 a.m. to 6:30 p.m.) at a temperature of 20–22 °C and humidity of 45–55% and were provided ad libitum access to food (Diet 4RF25 GLP; Mucedola, Settimo Milanese, Milan, Italy) and water. The experimental protocols performed in the present study were in accordance with the U.K. Animals (Scientific Procedures) Act of 1986 and associated guidelines and the new European Communities Council Directive of September 2010 (2010/63/EU). Experimental protocols were approved by the Italian Ministry of Health (license n. 335/2016-PR) and by the Animal Welfare Body of the University of Ferrara. According to the ARRIVE guidelines, all possible efforts were made to minimise the number of animals used, to minimise the animals’ pain and discomfort and to reduce the number of experimental subjects. For the overall study 84 mice were used. In the analysis of behavioural/histopathological responses for each treatment (vehicle, 6 different (±)*cis*-4,4′-DMAR doses (0.1, 1, 3, 10, 30 and 60 mg/kg), 2 different (±)*trans*-4,4′-DMAR doses (30 and 60 mg/kg), 3 different interaction of (±)*cis*-4,4′-DMAR (1, 10 and 60 mg/kg) with (±)*trans*-4,4′-DMAR (30 mg/kg)) 6 mice were used (total mice used: 72); in the behavioural/urine excretion studies for each treatment ((±)*cis*-4,4′-DMAR 10 mg/kg, (±)*trans*-4,4′-DMAR 10 mg/kg and (±)*cis*-4,4′-DMAR + (±)*trans*-4,4′-DMAR)) 4 mice were used (total mice used: 12).

### 4.2. Drug Preparation and Dose selection

*Cis* and *trans* 4,4′-DMAR were obtained from synthesis of 4-methylpropiophenone before they were monitored by existing legislation. In particular, the two isomers, (±)*cis*-4,4′-DMAR and (±)*trans*-4,4′-DMAR was synthesised as previously described by Brandt and co-workers [3]. Drugs were initially dissolved in absolute ethanol (final concentration was 2%) and Tween 80 (2%) and brought to the final volume with saline (0.9% NaCl). The solution made with ethanol, Tween 80 and saline was also used as the vehicle. These drugs were administered by intraperitoneal injection (i.p.) at a volume of 4 μL/g. The wide range of doses of (±)*cis*-4,4′-DMAR (0.1–60 mg/kg; i.p.) and (±)*trans*-4,4′-DMAR (30 and 60 mg/kg i.p.) were chosen based on interspecies dose scaling [105] in order to test in mice doses that corresponded to low, intermediate and high doses in humans (Table 5) [6,10,13].

The co-administration dosages (e.g., 1 mg/kg *cis*-4,4′-DMAR + 30 mg/kg *trans*-4,4′-DMAR, 10 mg/kg *cis*-4,4′-DMAR + 30 mg/kg *trans*-4,4′-DMAR and 60 mg/kg *cis*-4,4′-DMAR + 30 mg/kg *trans*-4,4′-DMAR; i.p.) were chosen based on results obtained in the dose-response curve of single enantiomers *cis* and *trans*.

### 4.3. Behavioural Studies: Physiological and Neuro-Behavioural Responses

The effects of (±)*cis*-4,4′-DMAR, (±)*trans*-4,4′-DMAR and their co-administration were investigated by a protocol widely used in studies of “safety pharmacology” for the preclinical characterization of new molecules in rodents [106,107,108,109]. This protocol includes a series of observational behavioural tests carried out in a consecutive manner that monitor the animal responses up to 5 h after compound injections. In the present study, we reported the observation of physiological (sweating, salivation, core temperature) and neuro-behavioural (detection of psychomotor agitation, spontaneous and stimulated aggressiveness, convulsions) changes occurring in mice after injection of (±)*cis*-4,4′-DMAR, (±)*trans*-4,4′-DMAR and their co-administration. Before pharmacological treatment the mouse was placed on a square plate (30 cm× 30 cm) raised from the ground (20 cm) and left free to move on the plate for 10 min (habituation period). The animal was then injected with the vehicle, substances alone ((±)*cis*-4,4′-DMAR or (±)*trans*-4,4′-DMAR) or in *cis + trans* co-administration and placed on the plate. From this moment, the recording time of the physiological and neuro-behavioural responses started, and these were detected in 5-min sessions.

After the first 5 min, the animal was removed from the plate; the body core temperature was then measured, and the aggression tests were performed. The 5-min observation period was repeated at 15–30–60–90–120–180–240–300. *Psychomotor agitation* in the mouse was not simply related to an increase in spontaneous locomotion itself, but by an “unusual hyperactivity” which was characterised by rapid and sudden horizontal displacements in all directions, sustained turning behaviour, involuntary falls from the high plate during the evaluation of the tests, stereotyped movements and rearing activity. Mouse *sweating and salivation* were reported only if present or not. The *core temperature* was determined using a probe (1 mm diameter) that was gently inserted after lubrication with liquid Vaseline into the rectum of the mouse (to about 2 cm) and left in position until the temperature stabilised (about 10 s) [106]. The probe was connected to a digital thermometer. *Spontaneous aggressive response* was measured based on the number of times a mouse bit a grey cloth put in front of its snout. During the test, each mouse was free to move within the cage. In the case of *stimulated aggressiveness*, each mouse was manually restrained and held in a supine position following which an object was brought near the mouth. For both spontaneous and stimulated aggressive behaviour tests, a grey cloth was placed in front of the nose of each mouse 10 consecutive times (score 0/10 not aggressive, score 10/10 very aggressive). *Convulsions* were defined as a loss of righting reflexes for at least 5 s, combined with the presence of body tremors and clonic or tonic limb movements [110,111]. *Deaths* caused by 4-4’-DMAR injection occurring during the observation period (5 h) were also reported as frequency (%) and time of death (min). Experiments were conducted inside the LARP (Centralised Preclinical Research Laboratory, University of Ferrara, Ferrara, Italy) in a thermo-stated (temperature 20–22 °C, humidity about 45–55%) and light controlled (about 150 lux) room, in which there was a background noise of about 40 ± 4 dB. The experiments, conducted in blind by trained observers working together in pairs [92], were videotaped and eventually analysed off-line by a different trained operator. The experiments were performed between 8:30 a.m. and 2:00 p.m.

### 4.4. Collection of Tissue Samples

Following the acute death or the sacrifice of the animals (after 5-h observation period by dislocation of the spine), brains were taken. These were fixed in 4% buffered formalin and then sampled and incorporated into the paraffin (Leica ASP300 processor, Histoline TEC2900 incorporator; Leica Microsystems Srl, Buccinasco, Milano, Italy). Subsequently, the cut (performed with the Leica HistoCore autocut microtome; Leica Microsystems Srl, Buccinasco, Milano, Italy) was designed to create sections with a thickness of 5 µm for each sample.

#### 4.4.1. Histological Procedure

For histological investigations, the sections were stained with haematoxylin/eosin and then observed under optical microscope (Nikon Eclipse E90i; Nikon, Roma, Italy).

#### 4.4.2. Immunohistochemical Procedure

On 5-µm-thick paraffined lung sections, we evaluated the expression of a panel of markers: heat shock proteins (HSP27, HSP70, HSP90), SMAC/DIABLO, NF-kB, iNOS, NOX-2, NT, 8-OHDG. Dilution of antibodies and pre-treatments necessary for antigen retrieval are shown in Table 6.

We used a detection system composed of a biotinylated secondary antibody and HRP-conjugated streptavidin (4plus HRP Universal Detection, Biocare Medical). 3′-,3Diaminobenzidine (DAB) and H2O2 (Betazoid DAB Chromogen Kit, Biocare Medical, Concord, CA, USA) were used as chromogen/substrate. The subsequent counterstaining with haematoxylin–eosin allowed the visualization of cell morphology and nuclei. Once this procedure was completed, the slides were mounted and observed under optical microscope. In the pictures (sections of the frontal cortex; Figure 3, Figure 4 and Figure 5) to evaluate the variation in the expression of markers (Smac, NOX2, iNOS, 8-OHdG, NT, HSP27, HSP70, HSP90) following treatments, the change in intensity of brown colour (staining of the immunohistochemical technique) in the control and treated tissue was measured. Quantification of Smac, NOX2, iNOS, 8-OHdG, NT, HSP27, HSP70 and HSP90 positive-stained areas was performed by the ImageJ software (imagej.nih.gov/ij/ Accessed date 12 May 2020). One image for each animal of the different experimental groups was processed. Positivity was expressed as an extension of the stained analysed area.

### 4.5. Excretion Studies on Urine Samples

#### 4.5.1. Chemicals and Reagents

*Cis*/*trans* 4,4′-DMAR (4-methyl-5-(4-methylphenyl)-4,5-dihydrooxazol-2-amine were synthetised and provided by the University of Ferrara Department of Organic Chemistry. Amphetamine-D11 employed as internal standard was purchased by Sigma–Aldrich (Milan, Italy). All chemicals (i.e., formic acid, acetic acid, ammonium formate, ammonium acetate, sodium phosphate, sodium hydrogen phosphate, potassium carbonate, potassium hydrogen carbonate, ethylacetate) were of analytical or HPLC grade and provided by Carlo Erba (Milan, Italy) and Sigma–Aldrich (Milan, Italy). The ultrapure water used was of Milli-Q grade (Millipore Italia, Vimodrone, Milan, Italy). The enzyme β-glucuronidase from *E. Coli* as well as the mixture β-glucuronidase/arylsulfatase (from *Helix pomatia*) was purchased from Roche (Monza, Italy).

#### 4.5.2. Dose and Sample Collection

To study the possible metabolic interaction between *cis* and *trans* isomers, we chose to evaluate urinary excretion and metabolic profile of 4,4′-DMAR, as this allowed us to study and correlate physiological and neuro-behavioural responses in the same individual. For the in vivo studies, three different groups of mice were selected. To the first group, a dose of 10 mg/kg of *cis-*4,4′-DMAR was administered. To the second group, a dose of 10 mg/kg of both *cis* and *trans-*4,4′-DMAR was administered. The third group was selected for the collection of urine blank samples. Urine samples were collected in the range of 0–6 h after 4,4′-DMAR or vehicle were intraperitoneally injected.

#### 4.5.3. Excretion Studies

Urine samples were treated and analysed through a protocol already used by our group for the analysis of the excretion of stimulant drugs in mice [84,112]. Briefly, our protocol allowed the conversion of conjugated metabolites (i.e., sulfo- and glucorono-conjugates) to phase I metabolites after two hydrolysis steps. The first employed β-glucuronidase for the hydrolysis of glucorono-conjugates (working pH of the enzyme mixture 7.4). The second used a mixture of β-glucuronidase/arylsulfatase for the hydrolysis of sulfo-conjugates (working pH of the enzyme mixture 5). Hydrolysed samples were extracted adding to the samples carbonate buffer (final pH 11) and 7 mL of ethyl acetate. The organic layer was next evaporated to dryness under nitrogen stream at a temperature of 30 °C. The residue was reconstituted with 50 μL of mobile phase and analysed with a targeted LC-MS/MS technique. In details, samples were analysed using an Agilent 1200 series HPLC instrument equipped with a SUPELCO Discovery C18 column (15 cm × 2.1 mm × 5 µm) coupled with an API4000 QqQ mass spectrometer (Sciex, 500 Old Connecticut Path, Framingham, MA, USA). The chromatographic and mass spectrometer parameters were described in the above cited articles.

### 4.6. Data and Statistical Analysis

*Sweating and salivation* were expressed as frequency (e.g., % of animals that developed symptoms); *psychomotor agitation* was expressed as frequency (% of animals) and duration (total time in mins). *Core temperature* values were expressed as the difference between control temperature (before injection) and temperature following drug administration (Δ°C). *Aggressiveness* was expressed as frequency (% of animals which become aggressive) and score (number of bites). Convulsions were expressed as frequency (% of animals that developed seizure), episodes (e.g., number of seizure events), latency of first episode (sec.) and duration of each episode (sec.). *Lethality* was expressed as frequency (% of animals that died during the observation period (5 h) or during the following 24 h). The statistical analysis of the effects of the individual substances in different concentrations were performed using a one-way ANOVA followed by a Bonferroni test for multiple comparisons. A Student’s t-test was used to determine statistical significance (*p* < 0.05) between two groups. Regarding immunohistochemical analysis, *positivity* was expressed as the extension of stained analysed area. Results were expressed as means ± mean standard error (SEM). Data were analysed by one-way analysis of variance (ANOVA), followed by Tukey’s post-hoc test. For all tests, a *p* value of <0.05 was considered statistically significant. All statistical analyses were performed using GraphPad Prism 8 software for Windows (La Jolla, CA, USA).

## 5. Conclusions

Indeed, consistent with current findings one would conclude that some of the stimulant NPS such as the 4,4′-DMAR molecules here thoroughly investigated could putatively exhibit very severe clinical toxicity levels. The clinical–toxicological relevance of the study is due to the use of doses of 4-4′-DMAR in mice that are equivalent to the dose of 4-4’-DMAR in humans, evoking mild, intermediate and strong behavioural and physiological responses (Table 5). The current drug scenario is changing very rapidly; the earliest of the possible appearance of a new substance might be evidenced on the deep, followed by a migration to the open, web. Eventually, the NPS would move into ‘head shops’ and/or the ‘street’ market, then reported by formal early warning systems, and new legislation would be implemented to counter the index NPS/substance [113]. Hence, an approach aiming at describing what is being discussed online by the web-based NPS enthusiasts ‘e-psychonauts’ [114] has been considered as potentially useful to identify in advance the NPS availability, market and diffusion [115]. To improve accuracy and provide a thorough evaluation of NPS pharmacology, further research should focus on an integrative model in which web-based analyses will be combined with more advanced research approaches. From this perspective, our currently ongoing related quantitative structure activity relationship (QSAR), docking and in silico studies will hopefully provide important findings in terms of which NPS, within a given class (e.g., stimulants; novel synthetic opioids; novel benzodiazepines) will present with higher levels of receptor affinities, and hence clinical potency. These data, taken from selected molecules, could then be used to plan further in vitro and in vivo/preclinical studies. Clinicians should be regularly informed about the range of NPS, their intake modalities, their psychoactive sought after-effects, the idiosyncratic psychotropics’ combinations, and finally their medical, psychobiological and psychopathological risks.

## Figures and Tables

**Figure 1 ijms-22-08771-f001:**
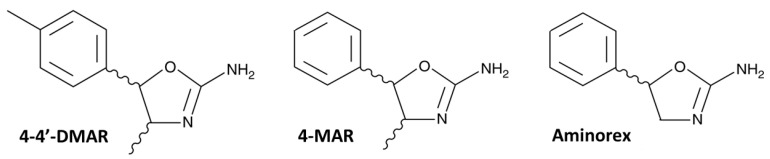
Structures of 4,4’-DMAR (4-Methyl-5-(4-methylphenyl)-4,5-dihydrooxazol-2-amine), 4-MAR (4-Methyl-5-phenyl-4,5-dihydrooxazol-2-amine) and Aminorex (5-phenyl-4,5-dihydro-1,3-oxazol-2-amine) copied from the Cayman Chemical website (https://www.caymanchem.com, accessed date 17 May 2021).

**Figure 2 ijms-22-08771-f002:**
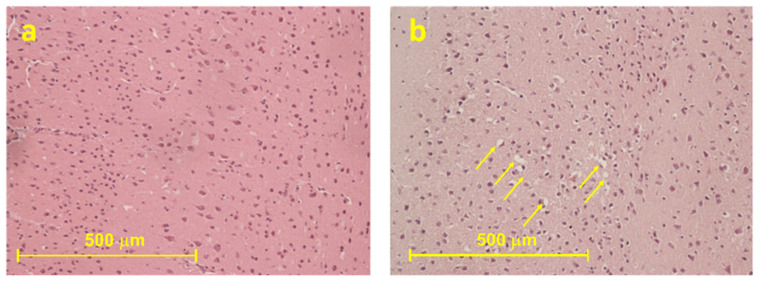
The frontal cortex of the treated mice ((±)*cis* 30; (**b**) showed mild edema (i.e., vacuolization areas are indicated by yellow arrows) compared to controls (**a**).

**Figure 3 ijms-22-08771-f003:**
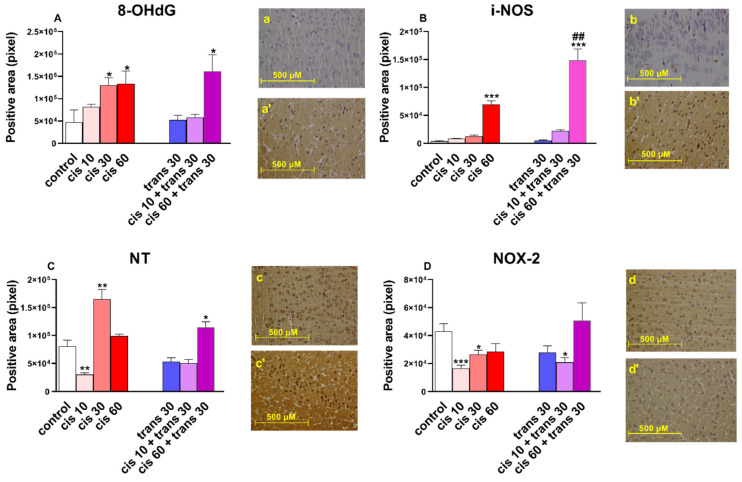
(**A**) Quantification and comparison of 8-OHdG positive areas in controls versus 4-4′-DMAR-mice ((±)*cis* 10, (±)*cis* 30, (±)*cis* 60, (±)*trans* 30, (±)*trans* 30 + (±)*cis* 10, (±)*trans* 30 + (±)*cis* 60). Representative image (light microscopy, 40×) of 8-OHdG immunostaining (brown areas) in the cortex of mice receiving 4-4′-DMAR ((±)*cis* 30, a’) and controls (a). (**B**) Quantification and comparison of iNOS positive areas in controls versus 4-4′-DMAR-mice ((±)*cis* 10, (±)*cis* 30, (±)*cis* 60, (±)*trans* 30, (±)*trans* 30 + (±)*cis* 10, (±)*trans* 30 + (±)*cis* 60). Comparison of (±)*cis* 60 versus (±)*trans* 30 + (±)*cis* 60. Representative image (light microscopy, 40×) of iNOS immunostaining (brown areas) in the cortex of mice receiving 4-4′-DMAR ((±)*cis* 60, b’) and controls (b). (**C**) Quantification and comparison of NT (nitrotyrosine) positive areas in controls versus 4-4′-DMAR-mice ((±)*cis* 10, (±)*cis* 30, (±)*cis* 60, (±)*trans* 30, (±)*trans* 30 + (±)*cis* 10, (±)*trans* 30 + (±)*cis* 60). Representative image (light microscopy, 40x) of NT immunostaining (brown areas) in the cortex of mice receiving 4-4′-DMAR ((±)*cis* 30, c’) and controls (c). (**D**) Quantification and comparison of NOX2 positive areas in controls versus 4-4′-DMAR-mice ((±)*cis* 10, (±)*cis* 30, (±)*cis* 60, (±)*trans* 30, (±)*trans* 30 + (±)*cis* 10, (±)*trans* 30 + (±)*cis* 60). Representative image (light microscopy, 40×) of NOX2 immunostaining (brown areas) in the cortex of mice receiving 4-4′-DMAR ((±)*cis* 30, d’) and controls (d). * *p* < 0.05, ** *p* < 0.01 and *** *p* < 0.001 different from control; ^##^
*p* < 0.001 different from (±)*cis* 60.

**Figure 4 ijms-22-08771-f004:**
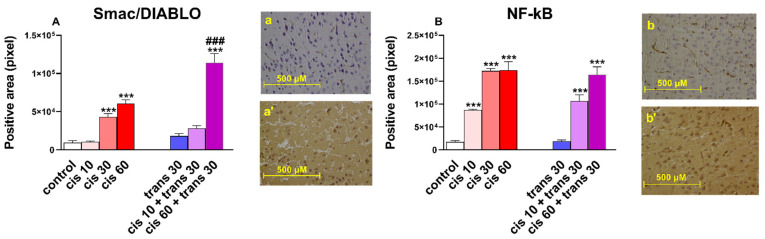
(**A**) Quantification and comparison of Smac/DIABLO positive areas in controls versus 4-4′-DMAR-mice ((±)*cis* 10, (±)*cis* 30, (±)*cis* 60, (±)*trans* 30, (±)*trans* 30 + (±)*cis* 10, (±)*trans* 30 + (±)*cis* 60).). Comparison of (±)*cis* 60 versus (±)*trans* 30 + (±)*cis* 60. Representative image (light microscopy, 40×) of Smac/DIABLO immunostaining (brown areas) in the cortex of mice receiving 4-4′-DMAR ((±)*trans* 30+(±)*cis* 60, a’) and controls (a). (**B**) Quantification and comparison of NF-κB positive areas in controls versus 4-4′-DMAR-mice ((±)*cis* 10, (±)*cis* 30, (±)*cis* 60, (±)*trans* 30, (±)*trans* 30 + (±)*cis* 10, (±)*trans* 30 + (±)*cis* 60). Representative image (light microscopy, 40x) of NF-κB immunostaining (brown areas) in the cortex of mice receiving 4-4′-DMAR ((±)*cis* 10, b’) and controls (b). *** *p* < 0.001 different from control; ^###^
*p* < 0.001 different from (±)*cis* 60.

**Figure 5 ijms-22-08771-f005:**
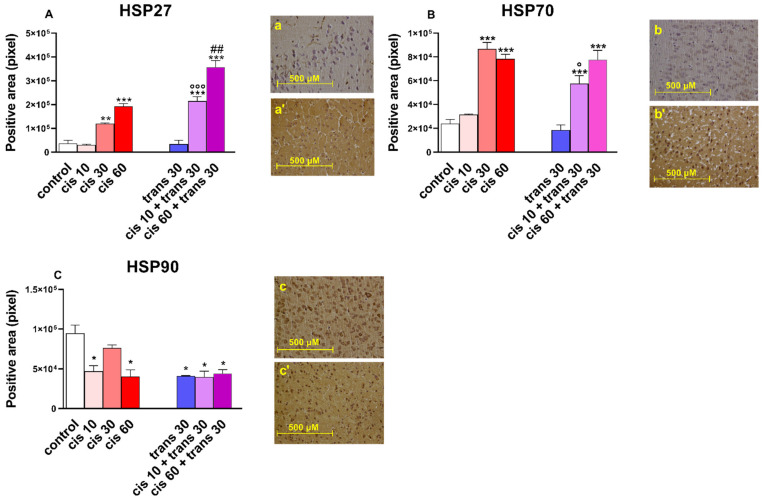
(**A**) Quantification and comparison of HSP27 positive areas in controls versus 4-4′-DMAR-mice ((±)*cis* 10, (±)*cis* 30, (±)*cis* 60, (±)*trans* 30, (±)*trans* 30 + (±)*cis* 10, (±)*trans* 30 + (±)*cis* 60). ). Comparison of (±)*cis* 60 versus (±)*trans* 30 + (±)*cis* 60). Comparison of (±)*cis* 10 versus (±)*trans* 30 + (±)*cis* 10 [°]. Representative image (light microscopy, 40×) of HSP27 immunostaining (brown areas) in the cortex of mice receiving 4-4′-DMAR ((±)*cis* 60, a’) and controls (a). (**B**) Quantification and comparison of HSP70 positive areas in controls versus 4-4′-DMAR-mice ((±)*cis* 10, (±)*cis* 30, (±)*cis* 60, (±)*trans* 30, (±)*trans* 30 + (±)*cis* 10, (±)*trans* 30 + (±)*cis* 60). Comparison of (±)*cis* 10 versus (±)*trans* 30 + (±)*cis* 10 [°]. Representative image (light microscopy, 40x) of HSP70 immunostaining (brown areas) in the cortex of mice receiving 4-4′-DMAR ((±)*cis* 60, b’) and controls (b). (**C**) Quantification and comparison of HSP90 positive areas in controls versus 4-4′-DMAR-mice ((±)*cis* 10, (±)*cis* 30, (±)*cis* 60, (±)*trans* 30, (±)*trans* 30 + (±)*cis* 10, (±)*trans* 30 + (±)*cis* 60). Representative image (light microscopy, 40×) of HSP90 immunostaining (brown areas) in the cortex of mice receiving 4-4′-DMAR ((±)*cis* 30, c’) and controls (c). * *p* < 0.05, ** *p* < 0.01 and *** *p* < 0.001 different from control; ° *p* < 0.05 and °°° *p*< 0.001 different from (±)*cis* 30; ^##^
*p* < 0.001 different from (±)*cis* 60.

**Figure 6 ijms-22-08771-f006:**
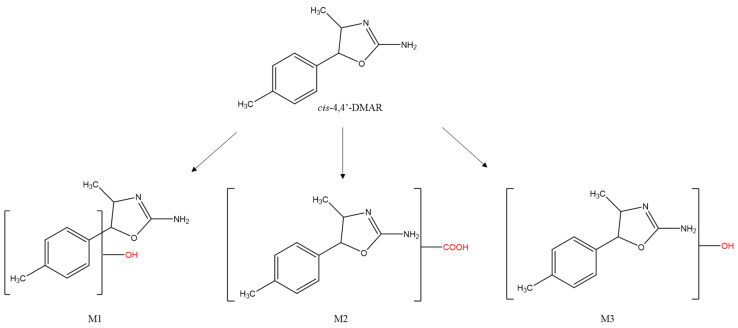
Structures of *cis*-4,4’-DMAR and its principal markers: hydroxylated (M1, M3) and carboxylated (M2).

**Figure 7 ijms-22-08771-f007:**
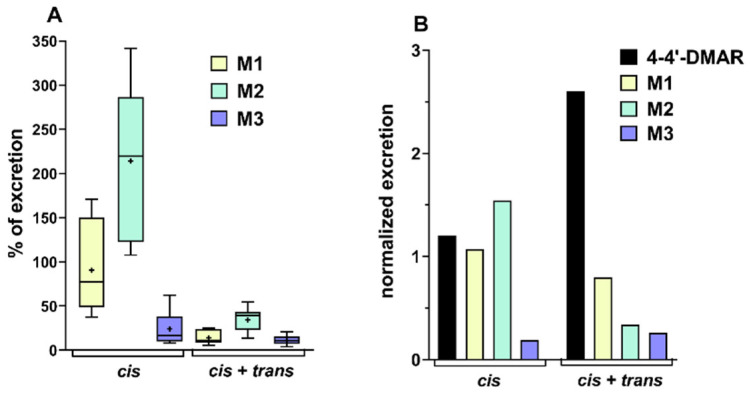
(**A**) Percentage ratio of excretion of M1, M2 and M3, the principal metabolites of *cis*-4,4′-DMAR for the administration of a dose of 10 mg/kg of 4,4′-DMAR (*cis*) and the same dose of both isomers (*cis* + *trans*). The box plots report maximum and minimum value, median and average (+). (**B**) Excretion data of 4,4′-*cis*-DMAR and its principal metabolites normalised to the sum of excretion of 4,4′-DMAR and its metabolites. Data normalised and reported, respectively, for the administration of cis or *cis* + *trans* isomers at a dose of 10 mg/kg.

**Table 1 ijms-22-08771-t001:** Effect of (±)*cis*-4,4′-DMAR (0.1–60 mg/kg i.p.) on neuro-behavioural changes (psychomotor agitation, aggressiveness, convulsions), physiological alterations (sweating, salivation, hyperthermia) and lethality in mice. The data here shown (see material and methods) refers to the mean ± SEM values relating to six animals for each treatment. The statistical analysis of the effects of the (±)*cis*-4,4′-DMAR in different concentrations were performed using a one-way ANOVA, followed by a Bonferroni test for multiple comparisons. A Student’s t-test was used to determine statistical significance (*p* < 0.05) between the two groups. ^a^
*p* < 0.05, versus cis 3 mg/kg; ^b^
*p* < 0.05, versus cis 10 mg/kg; ^c^
*p* < 0.05, versus vehicle; ^d^
*p* < 0.05, versus cis 30 mg/kg.

COMPOUND				*cis-4,4’-DMAR*					
Doses (mg/kg)			vehicle	0.1	1	3	10	30	60
**Psychomotor agitation**	*Frequency % (n° of mice)*		-	-	-	**100% (6)**	**100% (6)**	**100% (6)**	**100% (6)**
	*Duration (min)*		-	-	-	**54.2 ± 10.30**	**121.0 ± 12.35 ^a^**	**235.1 ± 15.25 ^ab^**	**280.1 ± 12.35 ^ab^**
	*Latency (min)*		-	-	-	**61.2 ± 6.30**	**28.2 ± 7.20 ^a^**	**3.2 ± 0.25 ^ab^**	**2.2 ± 0.15 ^ab^**
**Sweating**	*Frequency % (n° of mice)*		-	-	-	-	-	**100% (6)**	**100% (6)**
**Salivation**	*Frequency % (n° of mice)*		-	-	-	-	-	**100% (6)**	**50% (3)**
**Hyperthermia**	*Variation (Δ°C)*		**−0.28 ± 0.12**	**−0.18 ± 0.14**	**−0.22 ± 0.17**	**−0.34 ± 0.22**	**−0.41 ± 0.13**	**1.52 ± 0.11 ^c^**	**2.22 ± 0.12 ^c^**
	*Latency (min)*		-	-	-	-	-	**25 ± 0.25**	**7.12 ± 0.23 ^d^**
**Aggressiveness**	*Spontaneus*	*Frequency % (n° of mice)*	-	-	-	-	-	-	**nd**
	*Stimulated*	*Frequency % (n° of mice)*	-	-	-	-	**100% (6)**	**100% (6)**	**nd**
		*Score (n° of bites)*	-	-	-	-	**6 ± 0.23**	**10 ± 0.23 ^b^**	**nd**
**Convulsion**	*Frequency % (n° of mice)*		-	-	-	-	-	**50% (3)**	**100% (6)**
	*Episodes (n°)*		-	-	-	-	-	**3.5 ± 0.5**	**2.0 ± 0.41 ^d^**
	*Latency of first episode (sec)*		-	-	-	-	-	**10.5 ± 1.5**	**8.25 ± 1.1**
	*Duration of each episode (sec)*		-	-	-	-	-	**4.86 ± 1.26**	**16.14 ± 3.2 ^d^**
**Lethality**	*Frequency % (n° of mice)*		-	-	-	-	-	**50% (3)**	**100% (6)**
	*Time of death (min)*		-	-	-	-	-	**59.3 ± 2.3**	**31.9 ± 5.6 ^d^**
	*Surviving mice % (n° of mice)*		-	-	-	-	-	**50% (3)**	**0% (0)**

**Table 2 ijms-22-08771-t002:** Effect of (±)*trans*-4,4′-DMAR (30 and 60 mg/kg i.p.) on neuro-behavioural changes (psychomotor agitation, aggressiveness, convulsion), physiological alterations (sweating, salivation, hyperthermia) and lethality in mice. Data expressed (see material and methods) represents the mean ± SEM of six animals for each treatment. The statistical analysis of the effects of the (±)*trans*-4,4′-DMAR in different concentrations were performed using a one-way ANOVA followed by a Bonferroni test for multiple comparisons. A Student’s *t*-test was used to determine statistical significance (*p* < 0.05) between the two groups.

COMPOUND				*trans-4,4’-DMAR*	
Doses (mg/kg)			vehicle	30	60
**Psychomotor agitation**	*Frequency % (n° of mice)*		-	-	-
	*Duration (min)*		-	-	-
	*Latency (min)*		-	-	-
**Sweating**	*Frequency % (n° of mice)*		-	-	-
**Salivation**	*Frequency % (n° of mice)*		-	-	-
**Hyperthermia**	*Variation (Δ°C)*		**−0.28 ± 0.12**	**−0.37 ± 0.13**	**−0.35 ± 0.16**
	*Latency (min)*		-	-	-
**Aggressiveness**	*Spontaneus*	*Frequency % (n° of mice)*	-	-	-
	*Stimulated*	*Frequency % (n° of mice)*	-	-	-
		*Score (n° of bites)*	-	-	-
**Convulsion**	*Frequency % (n° of mice)*		-	-	-
	*Episodes (n°)*		-	-	-
	*Latency of first episode (sec)*		-	-	-
	*Duration of each episode (sec)*		-	-	-
**Lethality**	*Frequency % (n° of mice)*		-	-	-
	*Time of death (min)*		-	-	-

**Table 3 ijms-22-08771-t003:** Effect of co-administration of (±)*cis*-4,4′-DMAR (1, 10 and 60 mg/kg i.p.) and (±)*trans*-4,4′-DMAR (30 mg/kg i.p.) on neuro-behavioural changes (psychomotor agitation, aggressiveness, convulsion), physiological alterations (sweating, salivation, hyperthermia) and death in mice. Data expressed (see material and methods) represents the mean ± SEM of six animals for each treatment. The statistical analysis of the effects of the interactions between (±)*cis*-4,4′-DMAR and (±)*trans*-4,4′-DMAR were performed using a one-way ANOVA followed by a Bonferroni test for multiple comparisons. A Student’s *t*-test was used to determine statistical significance (*p* < 0.05) between the two groups. ^e^
*p* < 0.05, versus cis 10 mg/kg; ^f^
*p* < 0.05, versus cis 60 mg/kg.

COMPOUND				*cis-4,4’-DMAR*			*trans*	*(cis + trans)-4-4’DMAR*		
Doses (mg/kg)			vehicle	1	10	60	30	1 + 30	10 + 30	60 + 30
**Psychomotor** **agitation**	*Frequency % (n° of mice)*		-	-	**100% (6)**	**100% (6)**	-	**50% (3)**	**100% (6)**	**100% (6)**
	*Duration (min)*		-	-	**121.0 ± 12.35**	**220.1 ± 12.35**	-	**55.0 ± 5.0**	**185.5 ± 11.0 ^e^**	***nd***
	*Latency (min)*		-	-	**28.2 ± 7.20**	**2.2 ± 0.15**	-	**45.2 ± 5.20**	**6.2 ± 2.20 ^e^**	**0.45 ± 0.10 ^f^**
**Sweating**	*Frequency % (n° of mice)*		-	-	-	**100% (6)**	-	-	**33% (2)**	**100% (6)**
**Salivation**	*Frequency % (n° of mice)*		-	-	-	**50% (3)**	-	-	**33% (2)**	**50% (3)**
**Hyperthermia**	*Variation (Δ°C)*		**−0.28 ± 0.12**	**−0.22 ± 0.17**	**−0.41 ± 0.13**	**2.22 ± 0.12**	**−0.37 ± 0.13**	**−0.70 ± 0.15**	**1.6 ± 0.11 ^e^**	**2.25 ± 0.09**
	*Latency (min)*		-	-	-	**7.12 ± 0.23**	-	-	**28.0 ± 0.21**	**5.0 ± 0.23 ^f^**
**Aggressiveness**	*Spontaneus*	*Frequency % (n° of mice)*	-	-	-	***nd***	-	-	-	***nd***
	*Stimulated*	*Frequency % (n° of mice)*	-	-	**100% (6)**	***nd***	-	-	**100% (6)**	***nd***
		*Score (n° of bites)*	-	-	**6 ± 0.23**	***nd***	-	-	**10 ± 0.05 ^e^**	***nd***
**Convulsion**	*Frequency % (n° of mice)*		-	-	-	**100% (6)**	-	-	-	**100% (6)**
	*Episodes (n°)*		-	-	-	**2.0 ± 0.41**	-	-	-	**1.0 ± 0.0 ^f^**
	*Latency of first episode (sec)*		-	-	-	**8.25 ± 1.1**	-	-	-	**6.67 ± 2.73**
	*Duration of each episode (sec)*		-	-	-	**16.14 ± 6.2**	-	-	-	**40.0 ± 5.0 ^f^**
**Lethality**	*Frequency % (n° of mice)*		-	-	-	**100% (6)**	-	-	-	**100% (6)**
	*Time of death (min)*		-	-	-	**31.9 ± 5.6**	-	-	-	**11.0 ± 3.9 ^f^**
	*Surviving mice % (n° of mice)*		-	-	-	**0% (0)**	-	-	-	**0% (0)**

**Table 4 ijms-22-08771-t004:** Effect of co-administration of (±)*cis*-4,4′-DMAR (10 mg/kg i.p.) and (±)*trans*-4,4′-DMAR (10 mg/kg i.p.) on neuro-behavioural changes (psychomotor agitation, aggressiveness, convulsion), physiological alterations (sweating, salivation, hyperthermia) and death in mice. Data expressed (see material and methods) represents the mean ± SEM of four animals for each treatment. The statistical analysis of the effects of the interactions between (±)*cis*-4,4′-DMAR and (±)*trans*-4,4′-DMAR were performed using a one-way ANOVA followed by a Bonferroni test for multiple comparisons. A Student’s *t*-test was used to determine statistical significance (*p* < 0.05) between the two groups. ^e^
*p* < 0.05, versus cis 10 mg/kg.

COMPOUND			*cis-4,4’-DMAR*	*trans-4,4’-DMAR*	*(cis + trans)-4-4’DMAR*
Doses (mg/kg)			10	10	10+10
**Psychomotor** **agitation**	*Frequency % (n° of mice)*		**100% (4)**	-	**100% (4)**
	*Duration (min)*		**125.0 ± 10.12**	-	**171.0 ± 9.25 ^e^**
	*Latency (min)*		**26.4 ± 6.25**	-	**8.0 ± 4.25 ^e^**
**Sweating**	*Frequency % (n° of mice)*		-	-	-
**Salivation**	*Frequency % (n° of mice)*		-	-	-
**Hyperthermia**	*Variation (Δ°C)*		**−0.21 ± 0.15**	**−0.32 ± 0.1**	**1.58 ± 0.23 ^e^**
	*Latency (min)*		-	-	**32.0 ± 2.45**
**Aggressiveness**	*Spontaneus*	*Frequency % (n° of mice)*	-	-	-
	*Stimulated*	*Frequency % (n° of mice)*	**100% (4)**	-	**100% (4)**
		*Score (n° of bites)*	**5 ± 0.23**	-	**8 ± 0.13 ^e^**
**Convulsion**	*Frequency % (n° of mice)*		-	-	-
	*Episodes (n°)*		-	-	-
	*Latency of first episode (sec)*		-	-	-
	*Duration of each episode (sec)*		-	-	-
**Lethality**	*Frequency % (n° of mice)*		-	-	-
	*Time of death (min)*		-	-	-

**Table 5 ijms-22-08771-t005:** Correlation between mouse doses (mg/kg) and human equivalent doses (HED, mg/kg). The table also reported the correlation between doses and effects in human.

Mouse Dose (mg/kg)	HED	Human Dose	Human Dosage	Effects
	(mg/kg)	(mg)		
0.1	0.0081	0.486	***Low***	high state of vigilance, euphoria, decreased appetite, increased frequency of heartbeat and motor activity
1	0.081	4.86		
3	0.243	14.58		
10	0.81	48.6	***Intermediate***	restlessness, agitation and insomnia
30	2.43	145.8	***high***	involve severe anorexia, mild paranoia (sometimes hallucinations), hyperthermia, bruxism, facial spasms, an increase in aggression and desire for violence, seizures, an increased heart rate that will be involved in a cardiac arrest
60	4.86	291.6		

**Table 6 ijms-22-08771-t006:** Antibodies used for immunohistochemical analyses with the relative dilutions and antigenic retrieval methods.

Marker		Dilution	Retrieval
**HSP27**	Santa Cruz Biotechnology, Inc.	1:50	HIER (0.25 mM EDTA buffer )
**HSP70**	Santa Cruz Biotechnology, Inc.	1:50	HIER (0.25 mM EDTA buffer )
**HSP90**	Santa Cruz Biotechnology, Inc.	1:50	HIER (0.25 mM EDTA buffer )
**SMAC**	Santa Cruz Biotechnology, Inc.	1:100	HIER (0.01 M citrate buffer)
**NF-kB**	Santa Cruz Biotechnology, Inc.	1:50	HIER (0.25 mM EDTA buffer )
**iNOS**	Santa Cruz Biotechnology, Inc.	1:100	HIER (0.01 M citrate buffer)
**NOX-2**	Proteintech	1:100	HIER (0.01 M citrate buffer)
**NT**	Santa Cruz Biotechnology, Inc.	1:600	HIER (0.01 M citrate buffer)
**8OHDG**	Santa Cruz Biotechnology, Inc.	1:500	HIER (0.01 M citrate buffer)

## Data Availability

The data presented in this study are available on request from the first (Micaela Tirri) and corresponding author (Matteo Marti) for researchers of academic institutes who meet the criteria for access to the confidential data.

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
