# Peer review of "Worsening of the Toxic Effects of (±)Cis-4,4′-DMAR Following Its Co-Administration with (±)Trans-4,4′-DMAR: Neuro-Behavioural, Physiological, Immunohistochemical and Metabolic Studies in Mice"

_ijms, 2021, doi:10.3390/ijms22168771_

Round 1

Reviewer 1 Report

The authors discuss the toxic effects of a synthetic stimulant, 4,4’-DMAR.

Introduction

I think that the introduction section is long and could be shortened.

Result

2.1.1. The table 1 discusses the neuro-behavioral changes with 4-4DMAR.  The authors should add the number of animals used and the ones that survived the exposure. This should be added. Also what other observations did the authors find after exposure? I also thought that comparisons made in the table were difficult to follow (a vs b cvs d etc). Please check the grammar for the paragraphs

2.1.2 did the authors test the trans-4,4DMAR at the lower concentrations? Or are they suggesting that there is a dose-dependent effect only in higher concentrations? In that case, why is that? Please check the grammar for the paragraphs

2.2.1. I think it would be beneficial to the reader to know where the changes are happening in the picture. The authors should consider the addition of arrows or indications to indicate changes. The scale bar from the slides are missing. Please add.

2.2.2 The authors should add a line or two explaining the rationale behind choosing these markers.

Immunostaining bar graphs have been shown with differences. What are the comparison? Indicate in the legend for the figure. Please add what NT stands for in the legends. It is difficult to discern the

2.3 What was the reason for the authors to choose 10 over 30 mg/kg (since that the dose the effect has been observed?). Were there any differences with use of trans 4-4-DMAR alone?

Figure 4 and 5 : Could the authors show the difference with cis 30 or 60 for comparison in the figure? What are histological features altered with the cis 60 + trans 30? The authors need to add the figure and the explanation.

2.4 Previously, the authors used trans 30mg/kg for the experiments. Why was the concentration now changed to 10 mg/kg? Please add the explanation.

In all these experiments, were there any differences between the sexes with the exposure?

Discussion

The section should be shortened.

Explanations for the causes of psychomotor agitation, aggression should be cut down. Lines 419-29: although the references are cited what is the direct relation to the results? Please clarify.

Hyperthermia: the increase in ROS is described as an explanation for hyperthermia. Can these references be cited in relation to the results? I am finding it difficult to relate it back to the results.

Convulsions: The comparison to cocaine is described by the authors. Without a direct comparison in this animal cohort, this explanation is not pertinent and could be shortened.

What are the author's thoughts on the lack of response from the trans isomer? Please add

How does the literature compare to what the authors are showing? The authors should narrow the citations and description of the experiments. They should offer some of their thoughts on the possible mechanisms involved in 4-4-DMAR neurotoxicity.

Conclusions

The authors mention the relevance to clinical toxicity. Can the authors add a couple of sentences showing the relation between animal and human doses (in reference to table 5)?

Overall

The grammar should be checked thoroughly.  

If the statistics are presented with the F and P-value, perhaps consider using the APA presentation style. This would be helpful for the reader to determine the differences.

Line 448 please add the references?

Author Response

Response to Reviewer 1

We thank the Reviewer for his/her evaluation of our revised manuscript and for helpful concerns to improve the article.

In this revised version of the work we have address the major concerns of the referee (the parts highlighted in yellow have been included in the manuscript).

RevQ: Introduction: I think that the introduction section is long and could be shortened.

AA: As requested by the Reviewer the introduction was shortened

Result

RevQ: 2.1.1. The table 1 discusses the neuro-behavioral changes with 4-4DMAR. The authors should add the number of animals used and the ones that survived the exposure. This should be added. Also what other observations did the authors find after exposure? I also thought that comparisons made in the table were difficult to follow (a vs b cvs d etc). Please check the grammar for the paragraphs

AA: As suggested by Reviever we have modified the table 1 and inserted the number of mice (next to the frequency value in percentage) and a new row in table 1 showing how many mice survive. The caption of Table 1 indicates that 6 mice were used for each single dose of DMAR.

We made the same changes in the Table 3.

We thank the reviewer for the question. In this study we only performed these observations to avoid any other form of manipulation and possible stress to the animal.

We apologize for the letters describing the statistics, but it is also difficult for us to insert other symbols. We used the other symbols (asterisk, hash ...) to describe the statistics in the other figures with other statistical methods. We used letters to avoid confusion with the other statistical methods used.

As required by Reviewer the grammar was checked by a native speaker of the University of Hertfordshire, Hatfield, UK.

RevQ: 2.1.2 did the authors test the trans-4,4DMAR at the lower concentrations? Or are they suggesting that there is a dose-dependent effect only in higher concentrations? In that case, why is that? Please check the grammar for the paragraphs

AA: We agree with the Reviewer. We have also tested trans-4,4DMAR at lower doses (0.1 and 1 mg/kg). The compound is inactive. Therefore, we have reported only the highest doses tested (30 and 60 mg/kg) to highlight its inactivity even at high doses. Table 2 shows the trans-4,4DMAR at 30 and 60 mg/kg, while table 4 shows the dose of 10 mg/kg. We also included in the results (line 169) that trans-4,4DMAR was tested at lower doses: " The (±)trans-4,4’-DMAR tested at the lower doses of 0.1 and 1 mg/kg was inactive on all parameters studied."

RevQ:2.2.1. I think it would be beneficial to the reader to know where the changes are happening in the picture. The authors should consider the addition of arrows or indications to indicate changes. The scale bar from the slides are missing. Please add.

AA: We agree with the Reviewer and we modified the picture and the legend of figure 2. “(i.e. vacuolization areas are indicated by yellow arrows). We have modified the legend of figures 3, 4 and 5 and added the scale bar.

RevQ:2.2.2 The authors should add a line or two explaining the rationale behind choosing these markers.

AA: We clarified in the introduction the rationale for the choice of markers (line 122): “…Moreover, to highlight possible neurotoxic mechanisms we investigate the effect of 4,4’-DMAR on the expression of key markers of oxidative/nitrosative stress (8-OHdG, iNOS, NT and NOX2) and apoptosis (Smac/DIABLO and NF-κB), markers previously investigated in relation to MDMA, a stimulant with a pharmacological profile similar to 4-4’-DMAR. Furthermore, considering the preliminary data, which show hyperthermia, we evaluated the expression of heat shock proteins (HSP27, HSP70, HSP90), markers related to heat-induced response…”.

RevQ: Immunostaining bar graphs have been shown with differences. What are the comparison? Indicate in the legend for the figure. Please add what NT stands for in the legends. It is difficult to discern the

AA: We have modified the legend of figures 3, 4 and 5 according to the requests. The relevant statistics are reported in the text.

RevQ: 2.3 What was the reason for the authors to choose 10 over 30 mg/kg (since that the dose the effect has been observed?). Were there any differences with use of trans 4-4-DMAR alone?

AA: We used doses of cis 4-4-DMAR 10 mg/kg and trans 4-4-DMAR 10 mg/kg for the metabolic study because the combination of cis 4-4-DMAR 10 mg/kg + trans 4-4-DMAR 30 mg/kg caused marked behavioral effects including salivation and sweating in a percentage of animals. Since the excretion study is based on the collection of urine, we used a combination of doses that did not cause salivation and sweating. These responses could alter the amount of urine and the timing of excretion. The trans 4-4-DMAR 10 mg/kg alone did not produce any behavioural and metabolic (Chieffi, C.; Camuto, C.; De Giorgio, F.; De la Torre, X.; Diamanti, F.; Mazzarino, M.; Trapella C.; Marti, M.; Botrè, F. Meta-bolic profile of the synthetic drug 4,4′-dimethylaminorex in urine by LC–MS-based techniques: selection of the most suita-ble markers of its intake. Forensic Toxicology 2021, 39, 89-100, https://doi.org/10.1007/s11419-020-00544-9) effects.

RevQ: Figure 4 and 5 : Could the authors show the difference with cis 30 or 60 for comparison in the figure? What are histological features altered with the cis 60 + trans 30? The authors need to add the figure and the explanation.

AA:  The legends of figures 4 and 5 have been modified to be clearer.

RevQ: 2.4 Previously, the authors used trans 30 mg/kg for the experiments. Why was the concentration now changed to 10 mg/kg? Please add the explanation.

AA: The behavioural observations reported in Table 4 were made in the same animals from which urine was obtained for metabolic excretion studies. This has been done to correlate behavioral effects with actual levels of substance and metabolites in the urine excreted.

We reported in the text (line 353): “Animals from which urine for excretion studies were obtained (Fig.7) were simultaneously observed to see if changes in the excretion metabolism of the (±)cis-4,4’-DMAR stereoisomer were associated with changes in physiological and neuro-behavioural responses (Table 4)”.

RevQ: In all these experiments, were there any differences between the sexes with the exposure?

AA: As reported in materials and methods we used only male mice. The comparison of toxicity between males and females will be addressed in a further study.

Discussion

RevQ: The section should be shortened.

AA: We have corrected and shortened the discussion

RevQ: Explanations for the causes of psychomotor agitation, aggression should be cut down. Lines 419-29: although the references are cited what is the direct relation to the results? Please clarify.

AA: We agree with the Reviewer and we cut down the discussion related to psychomotor agitation and aggression. In order to reduce the discussion, we eliminate the sentences in lines 419-429. This part of the discussion wanted to explain the potential serotoninergic mechanism underlying the motor effects of 4.4-DMAR. However, this part is speculative and can be removed.

RevQ: Hyperthermia: the increase in ROS is described as an explanation for hyperthermia. Can these references be cited in relation to the results? I am finding it difficult to relate it back to the results.

AA: In the discussion the authors do not state that hyperthermia is caused by the increase in ROS. Conversely, hyperthermia can worsen the effects of radical stress.

RevQ: Convulsions: The comparison to cocaine is described by the authors. Without a direct comparison in this animal cohort, this explanation is not pertinent and could be shortened.

AA: We agree with the Reviewer. It is not appropriate to compare different behavioural studies with different strains of mice to obtain a quantitative comparison of 4,4’-DMAR versus cocaine effects …Therefore, we modified this discussion section and we eliminated the direct comparison with cocaine.

RevQ: What are the author's thoughts on the lack of response from the trans isomer? Please add

AA: This aspect is also very interesting for us. As we wrote in discussion the “(±)trans-4,4’-DMAR may have a different binding to plasma or tissue proteins and transporters that is common in chiral drug [Brocks, D.R. Drug disposition in three dimensions: an update on stereoselectivity in pharmacokinetics. Biopharm Drug Dis-pos. 2006, 27(8), 387-406, doi: 10.1002/bdd.517]. Its lack of pharmacological activity could therefore be due to the low availability of compound at the central level. Further studies will be undertaken to investigate this aspect.”

RevQ: How does the literature compare to what the authors are showing? The authors should narrow the citations and description of the experiments. They should offer some of their thoughts on the possible mechanisms involved in 4-4-DMAR neurotoxicity.

AA: Currently in the literature there are no studies on neurotoxicity from cis 4-4'-DMAR and not even from trans 4-4'-DMAR. This is the first study in which neurotoxicity, histological alterations (cerebral edema), changes related to oxidative stress and neuronal damage are demonstrated. By the mechanisms, based on the literature, and avoiding risky hypotheses we have hypothesized that hyperthermia is the main cause of neurotoxicity. Indeed, 4-4'-DMAR similarly to amphetamine- and methamphetamine-induced hyperthermia potentially enhances neurotoxicity through the disruption of protein function, ion channels and enhanced ROS production and through its effects on the vasculature [Bowyer, J.F.; Hanig, J.P., 2014]. The induced hyperthermia may lead to transitory breakdowns in the blood-brain barrier, which result in neurodegeneration and neuroinflammation in laboratory animals and brain pathology [Bowyer, J.F.; Hanig, J.P. Amphetamine- and methamphetamine-induced hyperthermia: Implications of the effects produced in brain vasculature and peripheral organs to forebrain neurotoxicity. Temperature (Austin). 2014, 1(3), 172-82, doi: 10.4161/23328940.2014.982049.; Sharma, H.S.; Sjöquist, P.O.; Ali, S.F. Drugs of abuse-induced hyperthermia, blood-brain barrier dysfunction and neurotoxi-city: neuroprotective effects of a new antioxidant compound H-290/51. Curr Pharm Des. 2007, 13(18), 1903-23, doi: 10.2174/138161207780858375.]

Conclusions

RevQ: The authors mention the relevance to clinical toxicity. Can the authors add a couple of sentences showing the relation between animal and human doses (in reference to table 5)?

AA: We thank the Reviewer for the suggestion. We add to the conclusion, line 752: ” The clinical-toxicological relevance of the study is due to the use of doses of 4-4'-DMAR in mice which are equivalent to the dose of 4-4'-DMAR in humans, evoking mild, intermedi-ate and strong behavioural and physiological responses (Table 5)”.

Overall

RevQ: The grammar should be checked thoroughly.

AA: As required by Reviewer the grammar was checked by a native speaker of the University of Hertfordshire, Hatfield, UK.

RevQ: If the statistics are presented with the F and P-value, perhaps consider using the APA presentation style. This would be helpful for the reader to determine the differences.

AA: As suggested by Reviewer we used the APA presentation style, and we modified the manuscript.

RevQ: Line 448 please add the references?

AA: We have deleted the sentence at Line 448 as the concept and correct reference were included in the previous sentence.

Reviewer 2 Report

The manuscript presents an interesting study regarding the neuro-toxicity of (±)cis-4,4’-DMAR following its co-administration with (±)trans-4,4’-DMAR in a murine model. The results of this study are very useful for the literature, investigating the effects of novel psychoactive substances with tremendous effects on consumer life.  I have some comments for the authors to improve their manuscript.

  1. Indicate what software was used to draw the chemical structures from Figure 1.
  2. Line 79, change “inhalation e oral administration” with “inhalation and oral administration”
  3. Check the manuscript for missing spaces between words, eg. see line 83: euphoriadecreased appetite
  4. Figure 3, 4 and 5 are not visible. You have to modify the font size and the format to can easily understand the legend of the graphs
  5. Please briefly explain the interspecies dose scaling used according to reference 127
  6. Please indicate how the chemicals were injected: intramuscular, intraperitoneal, etc
  7. Indicate for the sacrificed animals the method used.
  8. In the discussion section, the strengths and limitations of the study should be added.

Author Response

Response to Reviewer 2

We thank the Reviewer for his/her evaluation of our revised manuscript and for helpful concerns to improve the article.

In this revised version of the work we have address the major concerns of the referee (the parts highlighted in yellow have been included in the manuscript).

The manuscript presents an interesting study regarding the neuro-toxicity of (±)cis-4,4’-DMAR following its co-administration with (±)trans-4,4’-DMAR in a murine model. The results of this study are very useful for the literature, investigating the effects of novel psychoactive substances with tremendous effects on consumer life. I have some comments for the authors to improve their manuscript.

RevQ: Indicate what software was used to draw the chemical structures from Figure 1.

AA: For the preparation of Figure 1 we did not use any software. We simply copied the images from the Cayman Chemical website (https://www.caymanchem.com) and modified its through Powerpoint.

RevQ: Line 79, change “inhalation e oral administration” with “inhalation and oral administration”

AA: We have rectified the sentence (line 77).

RevQ: Check the manuscript for missing spaces between words, eg. see line 83: euphoriadecreased appetite

AA: We check the manuscript and arranged to add spaces between words (line 88).

RevQ: Figure 3, 4 and 5 are not visible. You have to modify the font size and the format to can easily understand the legend of the graphs

AA: We modified the graphs as requested by Reviewer.

RevQ: Please briefly explain the interspecies dose scaling used according to reference 127

AA: As reported by Reagan-Shaw and his colleagues, we simply multiplied the animal dose by 0.081 (values obtained by two constants: Animal (referred to mice) Km = 3 and Human Km = 37) and by the average weight of a healthy man (60 mg/kg).

RevQ: Please indicate how the chemicals were injected: intramuscular, intraperitoneal, etc

AA: We apologize for the forgetfulness. We only added the abbreviation in brackets next to the dosages (i.p.). We also added a sentence that explained the route of administration at line 614: “Drugs were administered by intraperitoneal injection (i.p.) at a volume of 4 μl/g”.

RevQ: Indicate for the sacrificed animals the method used.

AA: Survived mice of the trial, were sacrificed by dislocation of the spine because the use of CO2 could compromise the results sought through histopathological analysis of the cortex. We added it to line 671.

RevQ: In the discussion section, the strengths and limitations of the study should be added.

AA: As suggested by Reviewer in the discussion we stated that:” Overall, the strength of the present study related here to its clinical-toxicological relevance of the use of doses of 4-4'-DMAR in mice which are equivalent to those used in humans (HED; human equivalent dose); these were associated with mild, intermediate and strong responses (Tab. 5). Although the current study was solely based on the preclinical mouse model, this may allow a translational evaluation of the pharmaco-toxicological effects that could be observed in humans.” at line 383.

Round 2

Reviewer 1 Report

Figure 2: The scale for the pics are missing.

FIgure 3 Please add the scale to each micrographs. What is being compared in these pictures is not clear to the reader?

Why was there a consideration of male rats only? what is the rationale? If we are going to compare the levels to humans for clinical development, I the reviewers should add the rationale to the manuscript.

Author Response

We thank the Reviewer for his/her evaluation of our revised manuscript and for helpful concerns to improve the article.

In this revised version of the work we have address the major concerns of the referee (the parts highlighted in yellow have been included in the manuscript).

RevQ: Figure 2: The scale for the pics are missing.

AA: We apologize for this lack. As requested by the Reviewer the scale was inserted.

RevQ: FIgure 3 Please add the scale to each micrographs. What is being compared in these pictures is not clear to the reader?

AA: As requested by the Reviewer the scale was inserted to each micrographs.

We have specified in materials and methods that: ”In the pictures (sections of the frontal cortex; fig 3, 4 and 5) to evaluate the variation in the expression of markers (Smac, NOX2, iNOS, 8-OHdG, NT, HSP27, HSP70, HSP90) following treatments, the change in intensity of brown color (staining of the immunohistochemical technique) in the control and treated tissue was measured. Quantification of Smac, NOX2, iNOS, 8-OHdG, NT, HSP27, HSP70, HSP90 positive-stained areas was performed by the ImageJ software (imagej.nih.gov/ij/)..

RevQ: Why was there a consideration of male rats only? what is the rationale? If we are going to compare the levels to humans for clinical development, I the reviewers should add the rationale to the manuscript.

AA: The Reviewer's observation is very relevant and interesting. In the study we used only male mice to avoid at least in this phase further experimental variability given in female mice by hormonal variations in particular in the steroid hormone level oscillations during the estrus cycle. The present study is already complex due to the type of treatment carried out (administration of the cis 4-4'-DMAR compound at 6 different doses, of the trans 4-4'-DMAR compound at 2 different doses (over the lower doses 0.01-1 mg / kg), and of the co-administration of cis + trans and different doses) and the methods used (behavioral, immunohistochemical and evaluation of metabolite excretion). As future developments, a gender study will be performed in order to compare whether sex affects 4-4'-DMAR neurotoxicity.

In the discussion we stated that:” This study allows us to hypothesize that from a clinical point of view that an acute 4-4'-DMAR intoxication can be treated like that caused by other stimulants already known. Furthermore, the metabolic profile of 4-4'-DMAR excretion can cover an important aspect in clinical-toxicological and forensic investigations. While, changes in immunohistochemical markers (in particular HSPs) are indicative of metabolic changes in the brain that may be of help in forensic investigations to clarify, for example, the presence of hyperthermia in the brain and its possible pathophysiological relapse”.

Reviewer 2 Report

The authors addressed all my comments and concerns so the manuscript is ready for acceptance. 

Author Response

The authors addressed all my comments and concerns so the manuscript is ready for acceptance. 

We thank the reviewer for his/her advice and suggestions which allowed us to improve the manuscript